# Maternal obesity may disrupt offspring metabolism by inducing oocyte genome hyper-methylation via increased DNMTs

Shuo Chao[1†], Jun Lu[1†], Li-Jun Li[1], Hong-Yan Guo[1], Kuipeng Xu[2], Ning Wang[1], Shu-Xian Zhao[1], Xiao-Wen Jin[1], Shao-Ge Wang[1], Shen Yin[1], Wei Shen[1], Ming-Hui Zhao[1], Gui-An Huang[1], Qing-Yuan Sun[3]*, Zhao-Jia Ge[1]*

[1]College of Life Sciences, Institute of Reproductive Sciences, Key Laboratory of Animal Reproduction and Germplasm Enhancement in Universities of Shandong, Qingdao Agricultural University, Qingdao, China; [2]College of Horticulture, Qingdao Agricultural University, Qingdao, China; [3]Guangzhou Key Laboratory of Metabolic Diseases and Reproductive Health, Guangdong-Hong Kong Metabolism & Reproduction Joint Laboratory, Reproductive Medicine Center, Guangdong Second Provincial General Hospital, Guangzhou, China

**\*For correspondence:**
sunqy@gd2h.org.cn (Q-YS);
gejdssd313@163.com (Z-JiaG)

[†]These authors contributed equally to this work

**Competing interest:** The authors declare that no competing interests exist.

## eLife Assessment

This manuscript reports **important** findings on the impact of maternal obesity on offspring metabolism. It presents **solid** evidence that maternal obesity induces genomic methylation alterations in oocytes, which can be partly transmitted to F2 in females, and that melatonin is involved in regulating the hyper-methylation of high fat diet oocytes by increasing the expression of DNMTs via the cAMP/PKA/CREB pathway. This study would be of interest to biologists in the fields of epigenetics and metabolism.

**Abstract** Maternal obesity has deleterious effects on the process of establishing oocyte DNA methylation; yet the underlying mechanisms remain unclear. Here, we found that maternal obesity disrupted the genomic methylation of oocytes using a high-fat diet (HFD) induced mouse model, at least a part of which was transmitted to the F2 oocytes and livers via females. We further examined the metabolome of serum and found that the serum concentration of melatonin was reduced. Exogenous melatonin treatment significantly reduced the hyper-methylation of HFD oocytes, and the increased expression of DNMT3a and DNMT1 in HFD oocytes was also decreased. These suggest that melatonin may play a key role in the disrupted genomic methylation in the oocytes of obese mice. To address how melatonin regulates the expression of DNMTs, the function of melatonin was inhibited or activated upon oocytes. Results revealed that melatonin may regulate the expression of DNMTs via the cAMP/PKA/CREB pathway. These results suggest that maternal obesity induces genomic methylation alterations in oocytes, which can be partly transmitted to F2 in females, and that melatonin is involved in regulating the hyper-methylation of HFD oocytes by increasing the expression of DNMTs via the cAMP/PKA/CREB pathway.

## Introduction

Obesity has become a global health problem, affecting approximately 13% of the world's adult population, and over 340 million children and adolescents (WHO). This epidemic has profound implications

not only for reproductive health but also for the well-being of subsequent generations. Previous studies have demonstrated that maternal obesity reduces the function of the hypothalamic-pituitary-ovarian (HPO) axis (*Chen et al., 2022b*), deteriorates oocyte cytoplasmic quality and nuclear maturation, and disrupts genome methylation (*Broughton and Moley, 2017*). Reduced expression of *Stella* in oocytes induced by obesity results in global hypo-methylation in zygotes, which has an important contribution to the defective embryo development (*Han et al., 2018*). Furthermore, progeny of obese females has a higher risk of non-communicable diseases, such as obesity, diabetes, and cardiovascular diseases (*Godfrey et al., 2017*). Leptin is secreted from fat cells and associated with energy metabolism and obesity (*Obradovic et al., 2021*). Our previous study indicated that obesity altered the methylation status of *Leptin*, which might play a role in the metabolic disorders of female offspring. However, these methylation changes are not detected in the first-generation (F1) oocytes (*Ge et al., 2014*). The influence of maternal obesity on the genomic methylation of oocytes is still obscure. Thus, more studies are necessary to explore the role of DNA methylation in mediating the transgenerational transmission of metabolic syndrome induced by female obesity.

Obesity also perturbs glucose and lipid metabolism, which has negative effects on oocyte maturation and embryo development. Elevated levels of circulating free fatty acids in obese females contribute to oocyte lipotoxicity, while concomitantly diminishing mitochondrial function within oocytes (*Broughton and Moley, 2017*). The decreased melatonin levels are also reported in animals and humans by previous studies (*Overberg et al., 2022*; *Virto et al., 2018*). Melatonin has contributions to metabolism and DNA methylation in somatic cells (*Davoodvandi et al., 2022*). These findings indicate that the disturbed metabolism induced by obesity is closely linked to the compromised oocyte quality and aberrant methylation patterns. This study aims to elucidate the effects of obesity-induced metabolic changes on oocyte genomic methylation and its inhereditary implications in a murine model.

## Results

### Obesity alters the genomic methylation in oocytes

The obese mouse model was induced via a HFD (*Ge et al., 2014*; *Han et al., 2018*), and mice fed with a standard diet were used as a control (CD). The HFD group exhibited a significantly higher average body weight compared to the CD group (*Figure 1—figure supplement 1A, B*, B). Re-methylation in oocytes occurs in follicular development and is nearly complete at the germinal vesicle (GV) stage. The 5mC (5-methylcytosine) and 5hmC (5-hydroxymethylcytosine) levels in the GV oocytes of HFD mice were significantly higher than that in the CD group (*Figure 1A-C*). To further explore the effects of maternal obesity on the oocyte methylation, we examined the genomic methylation of metaphase II (MII) oocytes using whole-genome bisulfite sequencing for small samples (WGBS, Novogene, Beijing, China). The information on reads count, mapping rate, conversion rate, and sequencing depth was presented in *Supplementary file 1*. We found that global methylation in MII oocytes of the HFD group was higher than that in the CD group (*Figure 1D*). Methylated cytosine (C) can be classed into three types according to the context in the genome including CG, CHG, and CHH (H=A, T, or C). The methylated CG have more contributions to regulate gene expression. We found that the CG methylation level in MII oocytes of HFD was significantly higher than that in the CD group (*Figure 1E*). Differentially methylated CG is distributed across all chromosomes (*Figure 1—figure supplement 1C*). To further analyze the distribution of methylation, each functional region of genes was equally divided into 20 bins, and then the average methylation levels in the functional regions were calculated, respectively. CGIs (CG islands) and CGI shores were predicted using cpgIslandExt and repeat sequences were predicted using RepeatMasker. Results showed that the hyper-methylation was distributed in the promoter, exon, upstream 2 k, and downstream 2 k regions (*Figure 1F*, *Figure 1—figure supplement 1D*) of genes in HFD oocytes. These findings suggest that maternal obesity results in hyper-genome methylation in oocytes.

### Distribution of differentially methylated regions (DMRs)

We further analyzed the DMRs in oocytes, and identified 4340 DMRs between HFD and CD oocytes. These DMRs were defined by the following criteria: the number of CGs ≥4 and the absolute methylation difference ≥0.2. Among these, 2013 were hyper-DMRs (46.38%), and 2327 were hypo-DMRs

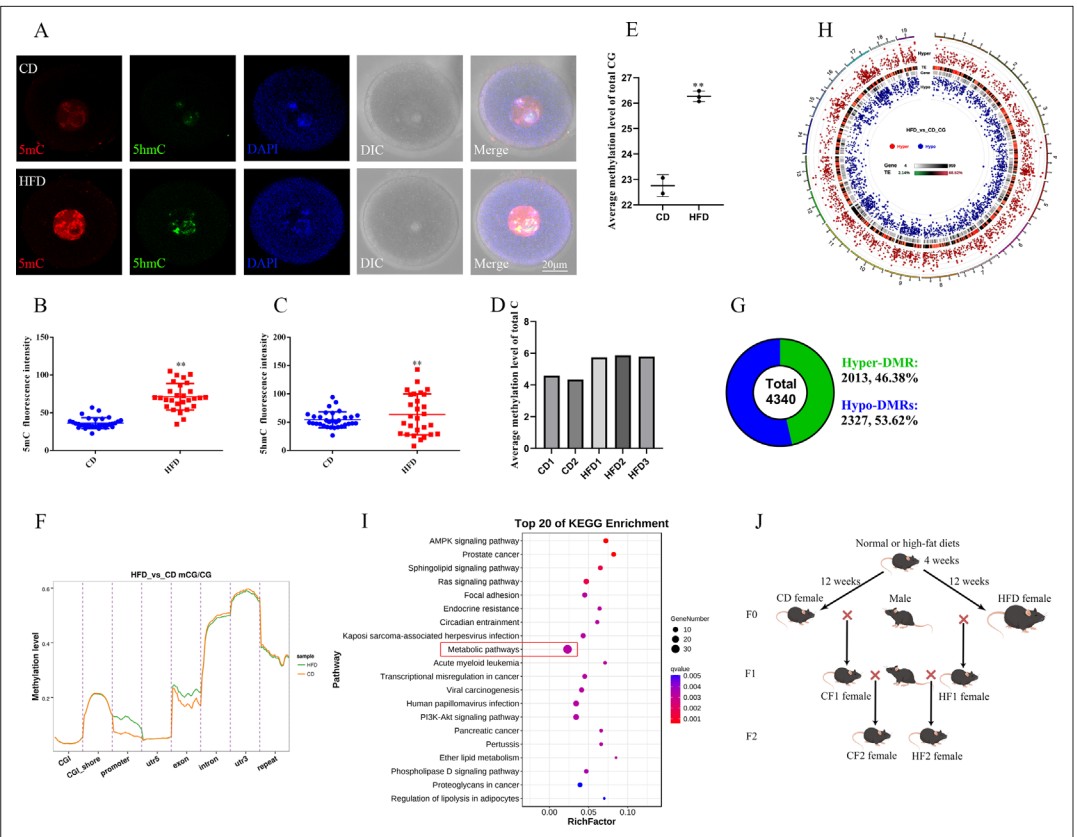

**Figure 1.** Maternal obesity alters the DNA methylation of oocytes. (**A**) Methylation levels of 5mC and 5hmC in oocytes (n>30). 5mC, 5-methylcytosine; 5hmC, 5-hydroxymethylcytosine; DAPI, chromatin. (**B, C**) Relative fluorescence intensity of 5mC and 5hmC in germinal vesicle (GV) oocytes. Data presented as mean ± SEM; two-tail t-test used, **p<0.01. (**D**) Genomic methylation level of MII oocytes examined by single-cell whole genome bisulfite sequencing. The control group (CD) has two replicates, and the obesity group (HFD) has three replicates. (**E**) Average genomic CG methylation level in MII oocytes. CD, control group; HFD, obesity group; data presented as mean ± SD, **p<0.01. (**F**) CG methylation levels at different regions in MII oocytes. CGI, CpG island; utr5, 5' untranslated region; utr3, 3' untranslated region; repeat, repeat sequence. (**G**) Total differentially methylated regions (DMRs) in oocytes of control and obesity groups. Hyper-DMRs, hypermethylated DMRs; hypo-DMRs, hypomethylated DMRs. (**H**) Distribution of DMRs on chromosomes in MII oocytes. Outside-to-in: chromosomes, hyper-DMRs, TEs (transcription end regions), and gene, hypo-DMRs. (**I**) Kyoto Encyclopedia of Genes and Genomes (KEGG) pathway enrichment of genes with DMRs at the promoter regions, and the top 20 pathways are presented. (**J**) Schedule of breeding. Female C57BL/6 mice fed with normal (CD) or high-fat diet (HFD) for 12 weeks were marked as F0. F1 was produced by F0 mated with normal males, respectively, and marked as CF1 and HF1; F2 was produced by female F1 mated with normal males and marked as CF2 and HF2, respectively.

The online version of this article includes the following figure supplement(s) for figure 1:

**Figure supplement 1.** Obese mouse model and DNA methylation in oocytes.

**Figure supplement 2.** dDifferentially methylated regions (DMRs) methylation at different regions.

**Figure supplement 3.** Distribution of the hypo- and hyper-differentially methylated regions (DMRs) in genomic elements.

---

(53.62%) (*Figure 1G*). These DMRs were distributed across all chromosomes (*Figure 1H*). We then annotated DMRs into different genomic regions including promoter, exon, intron, CGI, CGI shore, repeat, TSS (transcription start site), TES (transcription end site), UTR3 (3' end untranslated region), and UTR5 (5' end untranslated region) regions (*Figure 1—figure supplement 2A*), and the average methylation levels of DMRs in these regions were similar between HFD and CD oocytes (*Figure 1—figure supplement 2B*). However, the hypo-DMRs were enriched in the UTR3, repeat, and intron regions compared with hyper-DMRs (*Figure 1—figure supplement 3*).

Methylation level at promoters strongly contributes to the regulation of gene expression. We then analyzed the enrichment of genes with DMRs at promoters in KEGG (Kyoto Encyclopedia of Genes and Genomes) pathways using KOBAS online. Results indicated that the genes with DMRs at promoters were significantly enriched in metabolic pathways including amino acid metabolism pathways, carbohydrate metabolism pathways, lipid metabolism pathways, and metabolism of cofactors and vitamins pathways (*Figure 1I*, *Supplementary file 2*). A total of 35 genes with DMRs at promoters were included in metabolism pathways, 19 of these genes were with hyper-DMRs and 16 of these genes were with hypo-DMRs (*Supplementary file 3*). These results suggest that the altered methylation in oocytes induced by maternal obesity may play a role in the metabolic disorders in offspring.

## The disturbed methylation may be associated with the transgenerational inheritance of the metabolic disorders through females

Our recent study demonstrated that disturbed methylation in oocytes caused by uterine undernourishment can be partly transmitted to F2 oocytes via females, which may play a key role in the transgenerational inheritance of metabolic disorders (*Tang et al., 2023*). Here, we investigated the inheritance of altered methylation in HFD oocytes. F1 and F2 generations were produced as shown in the schedule in *Figure 1J*: HF1 and CF1, female HFD and CD, respectively, were mated with control males; HF2 and CF2, female HF1 and CF1, respectively, were mated with control males. We examined the glucose and insulin tolerance (GTT and ITT), and found that the GTTs and ITTs of F0, F1, and F2 females were impaired (*Figure 2A–C*). The inheritance of disrupted metabolism in females might be associated with the altered DNA methylation of oocytes. To address this question, we first examined the DNA methylation status of DMRs located at the promoters of *Bhlha15* (also known as *Mist1*, basic helix-loop-helix, a transcription factor), *Mgat1* (mannoside acetylglucosaminyltransferase 1), *Taok3* (serine/threonine-protein kinase 3), *Tkt* (transketolase), *Pik3cd* (phosphatidylinositol-4, 5-bisphosphate 3-kinase catalytic subunit delta), and *Pld1* (phospholipase D1) in the HFD and CD oocytes. We found that the methylation levels of hyper-DMRs including *Bhlha15*-DMR, *Mgat1*-DMR, and *Taok3*-DMR in HFD oocytes were significantly higher than in the CD group (*Figure 2D*). The methylation level of hypo-DMRs, *Pik3cd*-DMR in HFD oocytes was significantly lower than that in the CD group (*Figure 2D*). These results coincide with the genomic sequencing results. Whereas, the methylation level of *Tkt*-DMR (hypo-DMR) in HFD oocytes was higher than that in the CD group, and the methylation level of *Pld1*-DMR (hypo-DMR) was similar between the two groups, which contradicts the genomic sequencing results (*Figure 2—figure supplement 1A*). These findings suggest that some regions are false positives in genomic sequencing. To exclude the effects of somatic cell contamination, we examined the methylation level of paternally imprinted gene *H19*, which was low in both HFD and CD oocytes (*Figure 2—figure supplement 1B*). These findings indicate that the samples are not contaminated by somatic cells.

We then examined the methylation of DMRs in F1 livers using bisulfite sequencing (BS). Ten livers from five litters were analyzed for each group. Results revealed that the methylation levels of *Bhlha15*-DMR and *Mgat1*-DMR were higher and the methylation level of *Pik3cd*-DMR was lower in HF1 livers than that in CF1 livers (*Figure 2—figure supplement 2A–C*). The methylation level of *Tkt*-DMR in HF1 livers was lower than that in CF1 (*Figure 2—figure supplement 2D*), although it was higher in HFD oocytes compared with CD. This contradiction may be associated with the uterine environment of obesity and the reprogramming during early embryo development. We further examined the expression of several genes with DMRs at promoters, including hyper-methylated genes, such as *Bhlha15*, *Mgat1*, *Dgka*, *Pdpk1* and *Taok3*, and hypo-methylated genes, such as *Igf1*, *Map3k8*, *Pld1*, *Tkt*, *Pik3cd*, and *Sphk2*. The expression levels of *Bhlha15*, *Mgat1,* and *Pdpk1* were significantly lower and the expression levels of *Map3k8*, *Tkt*, *Pik3cd*, and *Sphk2* were significantly higher in HF1 livers compared to that in CF1 (*Figure 2—figure supplement 2E*, F). The expression trends of *Bhlha15*, *Mgat1*, *Tkt*, and *Pik3cd* were consistent with the methylation status at promoters in F1 livers. The expression of *Dgka*, *Taok3*, *Igf1*, and *Pld1* was not affected in F1 livers (*Figure 2—figure supplement 2E*, F). *Mgat1* is associated with lipid metabolism and obesity (*Jacobsson et al., 2012*; *Johansson et al., 2010*), *Tkt* regulates glucose metabolism (*Bartáková et al., 2016*; *Kang et al., 2018*), and *Pik3cd* is involved in lipid metabolism and diabetes (*Hood et al., 2019*; *Wójcik et al., 2014*). These

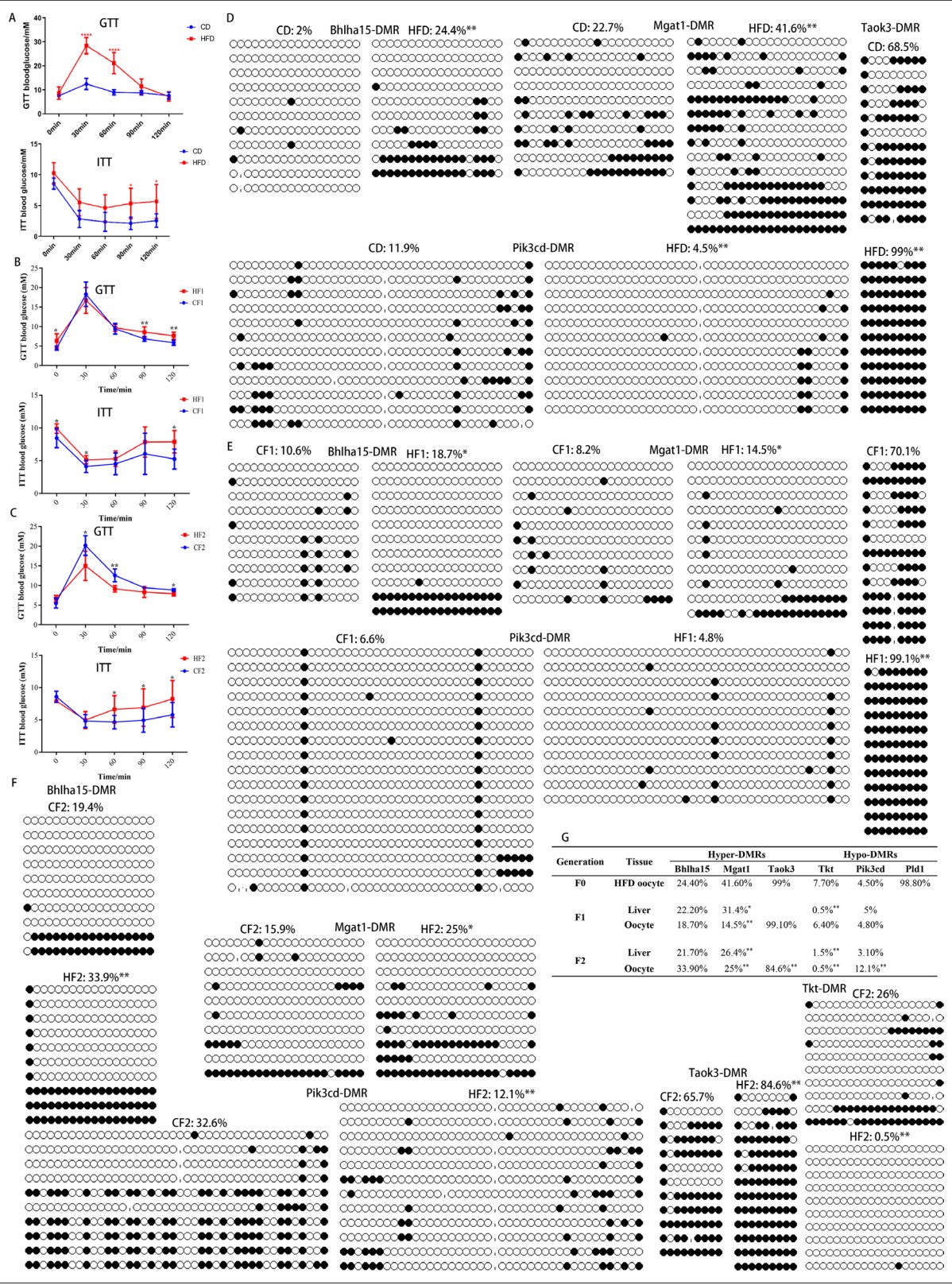

**Figure 2.** Transgenerational inheritance of metabolic disorders and altered DNA methylation. (**A-C**) Glucose tolerance (GTT) and insulin tolerance (ITT) were tested for female F0 (n: GTT, CD=6, HFD=11; ITT, CD=5, HFD=11), F1 (n: GTT, CF1=6, HF1=16; ITT, CH1=6, HF1=6), and F2 (n: GTT, CF2=5, HF2=5; ITT, CF2=7, HF2=5), respectively. * p<0.05; ** p<0.01. Data presented as mean ± SD; two-tail t test used. P value and n number are presented in the source data. (**D-F**) DMR methylation at the promoter regions of Bhlha15, Mgat1, Taok3, Tkt, and Pid3cd in F0, F1, and F2 oocytes was respectively

*Figure 2 continued on next page*

*Figure 2 continued*

examined using bisulfite sequencing. At least 10 available clones from 80-100 oocytes were used to calculate the methylation level. White circle, unmethylated CG; black circle, methylated CG. * p<0.05; ** p<0.01. Chi-square test used. (**G**) Inheritance of altered methylation in different generations was analyzed. * p<0.05; ** p<0.01. Chi-square test used.

The online version of this article includes the following source data and figure supplement(s) for figure 2:

**Source data 1.** Extended data for *Figure 2A, B, and C*.

**Figure supplement 1.** Methylation of H19 in oocytes.

**Figure supplement 2.** Methylation status of differentially methylated regions (DMRs) in F1 livers.

**Figure supplement 3.** Methylation level of Tkt-differentially methylated region (DMR) in F1 oocytes.

**Figure supplement 4.** Methylation of differentially methylated regions (DMRs) in F2 livers.

results suggest that the altered methylation in HFD oocytes is partly transmitted to F1 livers via oocytes, and that the abnormal methylation may be a reason for the disturbed metabolism of HF1.

If the altered methylation in HFD oocytes was inherited by HF1 oocytes, it would be transmitted to the F2 generation. Therefore, we examined the methylation of DMRs in F1 oocytes, and results revealed that *Bhlha15*-DMR, *Mgat1*-DMR, and *Taok3*-DMR were significantly hyper-methylated in HF1 oocytes compared with CF1 oocytes (*Figure 2E*). The methylation level of *Tkt*-DMR was significantly lower (*Figure 2—figure supplement 3*), and the methylation level of *Pik3cd*-DMR was slightly lower in the HF1 oocytes than that in the CF1 oocytes (*Figure 2E*). These results indicate that at least a part of the altered methylation in HFD oocytes is transmitted to F1 oocytes via females.

To confirm the transgenerational inheritance of the altered DNA methylation in HFD oocytes, we examined the methylation levels of DMRs in F2 livers. The methylation of *Bhlha15*-DMR and *Mgat1*-DMR was higher and the methylation level of *Pik3cd*-DMR was lower in HF2 livers than that in CF2 livers (*Figure 2—figure supplement 4A–C*). The methylation of *Tkt*-DMR in HF2 livers was similar to that in the CF2 group (*Figure 2—figure supplement 4D*). In addition, the expression levels of *Bhlha15*, *Pdpk1*, and *Mgat1* were significantly lower (*Figure 2—figure supplement 4E*), and the expression levels of *Pld1*, *Pik3cd*, and *Sphk2* were significantly higher in HF2 livers than that in CF2 (*Figure 2—figure supplement 4F*). The expression of the other gene was similar in the livers between HF2 and CF2 (*Figure 2—figure supplement 4E, F*, F). These results suggest that at least a part of the altered methylation in HF2 livers may be inherited from HFD oocytes, and that this alteration may be associated with the disrupted metabolism in F2 offspring.

We then examined the methylation of DMRs in F2 oocytes using BS, and found that the hyper-methylation of *Bhlha15*-DMR, *Mgat1*-DMR, and *Taok3*-DMR, and the hypo-methylation of *Pik3cd*-DMR and *Tkt*-DMR were maintained in HF2 oocytes (*Figure 2F*).

To better understand the inheritance of the altered methylation, we analyzed the methylation level of DMRs among generations compared with that in HFD oocytes (*Figure 2G*). For hyper-DMRs, the methylation of *Bhlha15*-DMR was maintained from HFD oocytes to HF2 oocytes. Compared to those in HFD oocytes, the methylation levels of *Mgat1*-DMR and *Taok3*-DMR were partly transmitted to HF2 oocytes. For hypo-DMRs, the methylation levels of *Tkt*-DMR and *Pik3cd*-DMR in HFD oocytes were inherited by HF2 (*Figure 2G*). We did not examine the methylation level of *Pld1*-DMR in F1 and F2 because it was similar in oocytes between CD and HFD (*Figure 2G*). These results suggest that only a part of the altered methylation in HFD oocytes can be transmitted to F2 oocytes, and that disrupted methylation may be a reason for the inheritance of the metabolic disorders in F1 and F2.

## Obesity alters the metabolomics of serum

Obesity alters the metabolism of glucose, fatty acids, and amino acids, which are essential for oogenesis. Thus, we suppose the altered metabolism may play a key role in the disturbed global methylation in HFD oocytes. We examined the metabolomics of serum using non-targeted approaches (BGI, Wuhan, China). We used LC-MS/MS to identify the variation in metabolites, including amino acids, carbohydrates, lipids, and phenols. The principal component analysis (PCA) showed that the PC1 and PC2, respectively, explained 44.6% and 9.96% of the total metabolite variation, respectively (*Figure 3A*). The distribution of extracts between groups was distinguishable (*Figure 3A*). We identified 538 differential metabolites based on the PLS-DA (robust orthogonal partial least squares-discriminant analysis) and T-test analysis with the following criteria: VIP

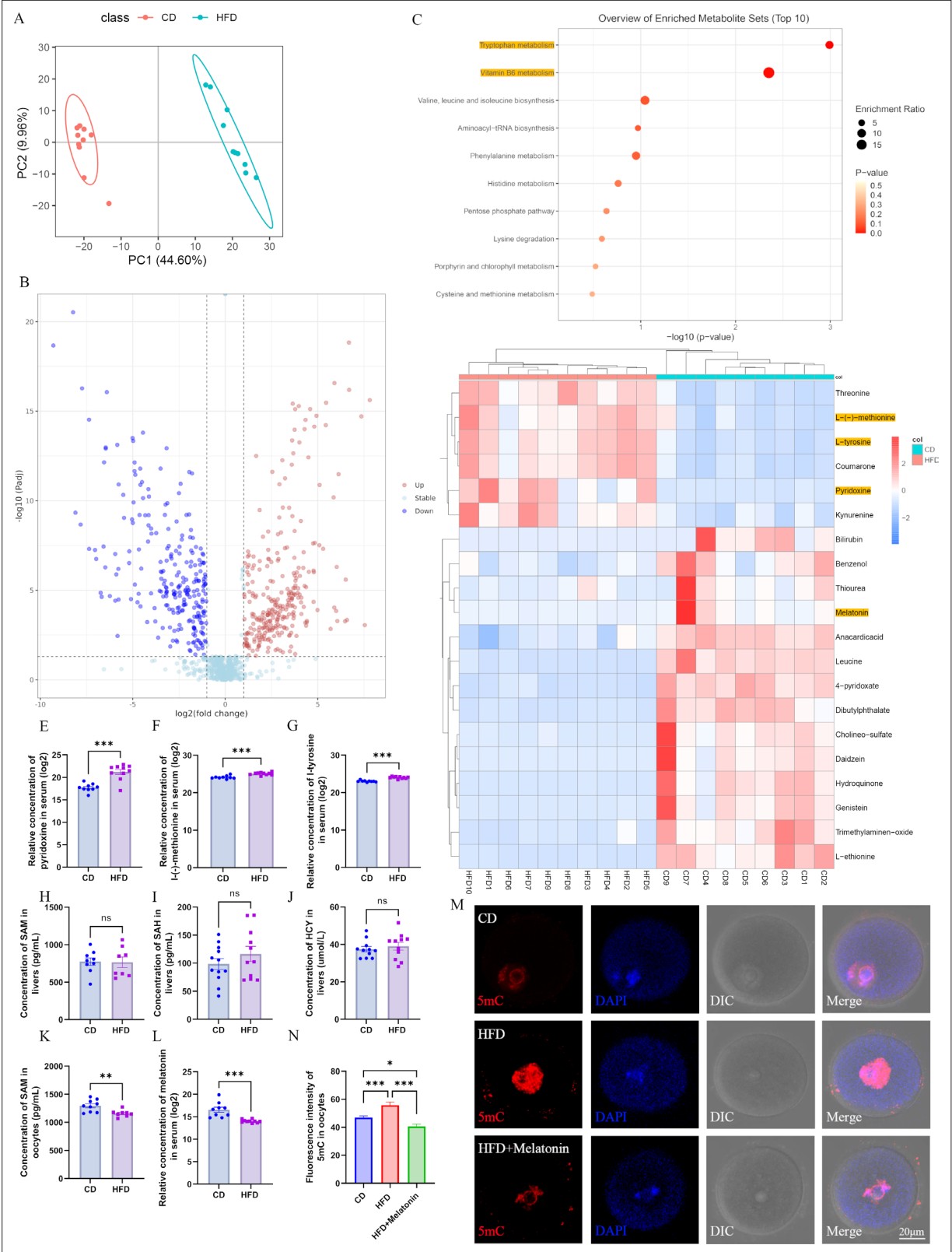

**Figure 3.** Maternal obesity alters the metabolome of serum. (**A**) Principal component analysis in control group (CD) and high-fat diet (HFD) mice. (**B**) Differential metabolites in the HFD serum compared with those in the CD group. Red circles, upregulated metabolites; blue circles, downregulated metabolites. (**C**) The enrichment of differential metabolites was analyzed using Kyoto Encyclopedia of Genes and Genomes (KEGG), and the top 10 enrichment terms were presented. (**D**) Heat map of the top 20 differential metabolites in HFD serum. (**E–G**) Comparison of the concentrations of

*Figure 3 continued on next page*

*Figure 3 continued*

pyridoxine (n: CD=9, HFD=10; p=5.295×10$^{-5}$), methionine (n: CD=9, HFD=10; p=5.5×10$^{-5}$), and tyrosine (n: CD=9, HFD=10; p=1.532×10$^{-7}$) among the groups. *p<0.05; **p<0.01; ***p<0.001. Data presented as mean ± SEM; a two-tail t-test used. (**H–J**) Concentrations of S-adenosyl methionine (SAM) (n: CD=9, HFD=8; p=0.925814), S-adenosyl homocysteine (SAH) (n: CD=12, HFD=11; p=0.279946) and homocysteine (HCY) (n: CD=11, HFD=11; p=0.540962) in the livers were examined by ELISA. Ns, there was no statistical significance between groups. Data presented as mean ± SEM; a two-tail t-test used. (**K**) The concentration of SAM in oocytes was analyzed using ELISA. **p<0.01. Data presented as mean ± SEM; a two-tail t-test was used; n: CD=9, HFD=8; p=0.006335. (**L**) Relative concentration of melatonin in the serum. ***p<0.001. Data presented as mean ± SEM; a two-tail t-test was used; n: CD=9, HFD=10; p=0.00022. (**M**) Genomic DNA methylation in oocytes was examined using immunofluorescence. CD, control group; HFD, obesity group; HFD + melatonin, obese mice were treated with exogenous melatonin for 14 d. (**N**) Relative fluorescence intensity of 5mC was examined using Image J (CD, n=109; HFD, n=104, p=0.000639; HFD + melatonin, n=96, p=2.657×10$^{-7}$). *p<0.05; ***p<0.001. Data presented as mean ± SEM; a two-tail t-test was used. Source data are presented in *Figure 3—source data 1*.

The online version of this article includes the following source data and figure supplement(s) for figure 3:

**Source data 1.** Extended data for *Figure 3F-L, and N*.

**Figure supplement 1.** Concentrations of genistein and dibutylthalate in the serum of high-fat diet (HFD) and control group (CD).

(variable importance in projection) ≥1, fold change ≥1.2 or≤0.83, and *P*-value <0.05, including 288 upregulated and 250 downregulated metabolites (*Figure 3B*). The enrichment of differential metabolites was analyzed using KEGG, which revealed that differential metabolites were significantly enriched in tryptophan and vitamin B6 metabolism (*Figure 3C*). The top 20 differential metabolites are presented in *Figure 3D*. These results suggest that obesity disturbs the metabolomics of serum.

## Melatonin may play a key role in the genomic hyper-methylation of HFD oocytes

To investigate the associations between the metabolites and methylation in oocytes, we identified several differential metabolites in HFD mice, including pyridoxine (vitamin B6), L-methionine, melatonin, and L-tyrosine, which may be associated with the hyper-methylation of oocytes. The concentration of pyridoxine (vitamin B6), L-methionine, and L-tyrosine was significantly increased in HFD serum compared with the CD group (*Figure 3D–G*). During the DNA methylation process, the methyl group donor S-adenosyl methionine (SAM) is degraded into S-adenosyl homocysteine (SAH) and homocysteine (HCY) (*Chen et al., 2004*), and vitamin B6 serves as a co-factor (*Vaccaro and Naser, 2021*). Excessive methionine and vitamin B6 intake results in hyper-methylation (*Waterland, 2006*). These findings indicate that the hyper-methylation in HFD oocytes may be associated with the increased concentrations of methionine and pyridoxine. To confirm this hypothesis, we examined the concentrations of SAM, SAH, and HCY in livers and oocytes. Results showed that the concentrations of SAM, SAH, and HCY in HFD livers were similar to those in CD livers (*Figure 3H–J*), and the concentration of SAM in HFD oocytes was lower than that in the CD group (*Figure 3K*). These results suggest that the higher concentrations of methionine and pyridoxine in the serum of HFD mice may not be the main reason for the genomic hyper-methylation in oocytes.

As presented in *Figure 3D*, it is curious that the abundance of genistein, daidzein, and dibutylphthalate were also altered in HFD serum compared with those in CD, which might have contributed to the altered DNA methylation in the oocytes. These metabolites might be from the diets or materials used to collect samples. To confirm these results, we examined the concentrations of genistein and dibutylpthalate using ELISA. The results revealed that the concentrations of these metabolites were similar between HFD and CD serum (*Figure 3—figure supplement 1*), which suggests that these metabolites may have no effect on the altered methylation in oocytes.

As presented in *Figure 3D and L*, the concentration of melatonin in the HFD serum was significantly lower than that in the CD group. Low concentrations of melatonin have also been reported in obese rats (*Virto et al., 2018*) and humans (*Overberg et al., 2022*). Melatonin affects DNA re-methylation in oocytes (*Lan et al., 2018*; *Saeedabadi et al., 2018*; *Xiao et al., 2019*). Here, we found that if HFD mice were treated with exogenous melatonin for 14 d, the genomic hyper-methylation in HFD oocytes was significantly reduced (*Figure 3M and N*). These results suggest that the reduced melatonin concentration may be involved in regulating the hyper-methylation of HFD oocytes.

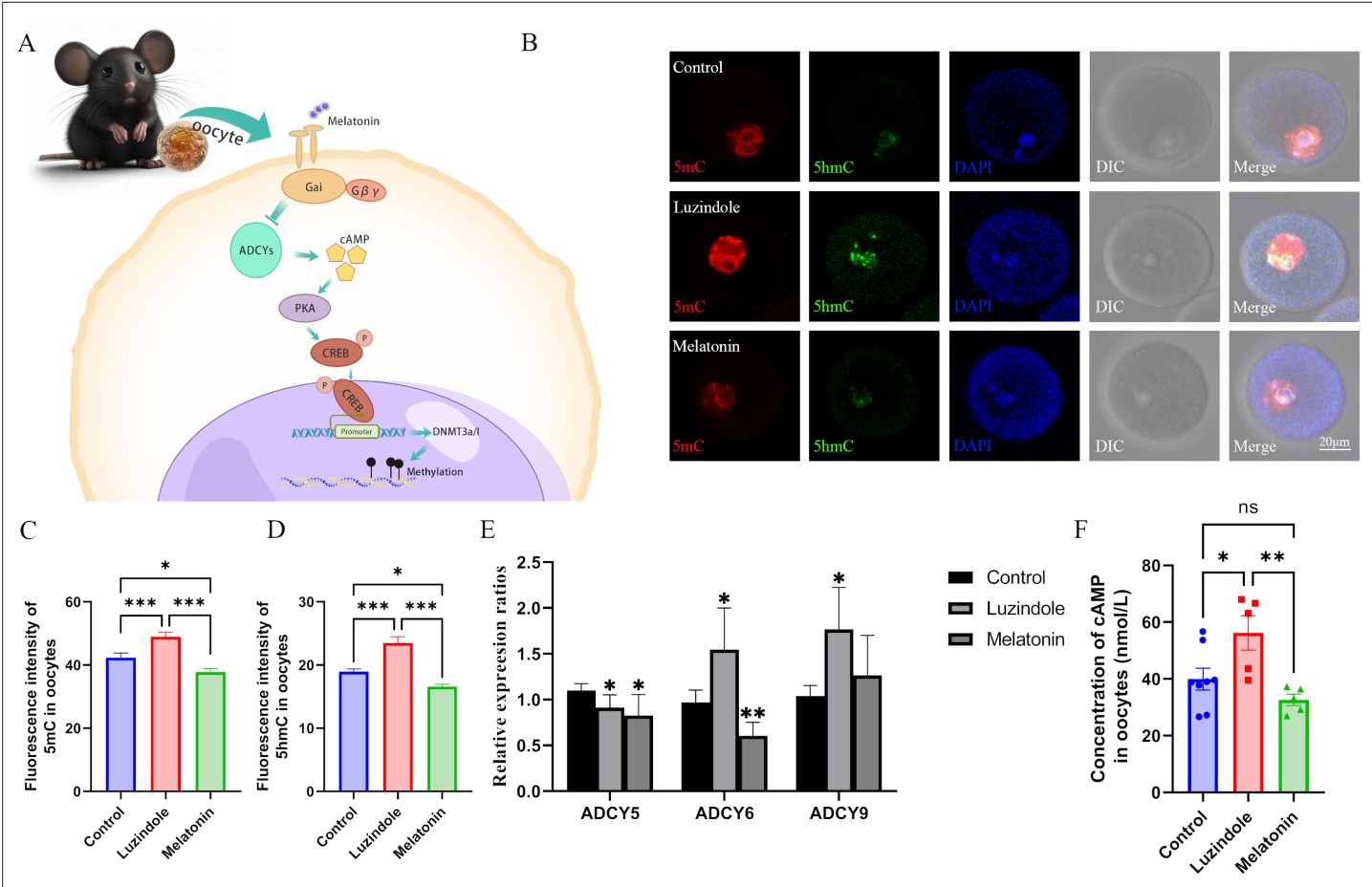

**Figure 4.** Melatonin regulates DNA methylation in oocytes. (**A**) Schedule of the possible pathway by which melatonin regulates DNA methylation in oocytes. According to previous studies, we predicted that melatonin might regulate DNA methylation in oocytes via the cAMP/PKA/CREB pathway. (**B**) Effects of melatonin and its inhibitor luzindole on oocyte methylation were examined using immunofluorescence. (**C, D**) The relative fluorescence intensities of 5mC and 5hmC were analyzed using Image J (5mC: Control, n=81; Luzindole, n=83, p=0.000886; Melatonin, n=86, p=5.19×10⁻¹⁰; 5hmC: Control, n=64; Luzindole, n=58, p=3.258×10⁻⁵; Melatonin, n=49, p=2.065×10⁻⁸). *p<0.05; ***p<0.001. Data presented as mean ± SEM; two-tail t-test was used. (**E**) The effects of melatonin and its inhibitor luzindole on the expression of adenylate cyclase (ADCY) in oocytes were examined by qPCR. *p<0.05; **p<0.01. Data presented as mean ± SD; two-tail t-test was used. p-value presented in *Figure 4—source data 1*. (**F**) The concentration of cAMP in oocytes was examined by ELISA. *p<0.05; **p<0.01. Data presented as mean ± SEM; a two-tail t-test was used; n: control=8, luzindole=5, p=0.035046, melatonin=5, p=0.006113. Source data are presented in *Figure 4—source data 1*.

The online version of this article includes the following source data and figure supplement(s) for figure 4:

**Source data 1.** Extended data for *Figure 4C-F*.

**Figure supplement 1.** Melatonin level in the blood and ADCYs expression in oocytes.

## Melatonin regulates genomic methylation of oocytes by increasing the expression of DNMTs via the cAMP/PKA/CREB pathway

Melatonin receptors (MT1 and 2), which are coupled with inhibitor G-protein (Gi), have been identified in oocytes and granulosa cells (*Jin et al., 2022*; *Wang et al., 2021*). Activated Gi inhibits the activation of adenylyl cyclases (ADCYs), resulting in a decrease of cAMP (cyclic adenosine monophosphate), which regulates the activation of protein kinase A (PKA) and CREB (cAMP response element-binding protein) (*Wongprayoon and Govitrapong, 2021*). Elevated cAMP level increases the expression of DNMTs resulting in hyper-methylation in HL-1 cardiomyocytes (*Fang et al., 2015*). We thus supposed that melatonin may regulate genomic methylation in oocytes via increasing the expression of DNMTs through the cAMP/PKA/CREB pathway (*Figure 4A*). To confirm this hypothesis, female C57BL/6 mice fed with a normal diet were treated with luzindole, an inhibitor of melatonin receptor, and the global methylation of 5mC and 5hmC was significantly increased in oocytes (*Figure 4B–D*). Luzindole did

not affect the concentration of melatonin in the serum (*Figure 4—figure supplement 1A*). However, excessive melatonin treatment significantly increased the concentration of melatonin in the serum (*Figure 4—figure supplement 1A*) and decreased the methylation levels of 5mC and 5hmC in oocytes (*Figure 4B–D*). These findings indicate that melatonin may regulate the re-methylation process in oocytes.

To confirm whether melatonin regulates methylation in oocytes via the cAMP pathway, we examined the expression of ADCYs in oocytes using RT-PCR, and found that ADCY5, 6, and 9 were expressed in oocytes (*Figure 4—figure supplement 1B*). The melatonin antagonist luzindole significantly increased the expression of ADCY6 and ADCY9 in oocytes, and melatonin reduced the expression of ADCY6 (*Figure 4E*). However, the expression of ADCY5 was lower in the luzindole and melatonin groups compared with the control (*Figure 4E*). In addition, the melatonin antagonist luzindole increased, while melatonin decreased the concentration of cAMP in oocytes, respectively (*Figure 4F*). These results suggest that melatonin may regulate the synthesis of cAMP via ADCY6 in oocytes. To further confirm the role of cAMP in regulating the methylation of oocytes, we treated mice with SQ22536, an inhibitor of ADCYs, and found that this treatment significantly reduced the global methylation of 5mC and the concentration of cAMP in oocytes (*Figure 5A–C*). Whereas, the ADCYs activator forskolin significantly increased the cAMP concentration and global methylation of 5mC in oocytes (*Figure 5A–C*). 8-Bromo-cAMP, a cAMP analogue, also increased the global methylation in oocytes (*Figure 5D and E*). cAMP functions by activating the downstream protein PKA. When we treated mice with H89 2HCL, a PKA antagonist, the global methylation of 5mC was significantly reduced in oocytes (*Figure 5F and G*). These results suggest that melatonin may mediate DNA methylation in oocytes via the cAMP/PKA pathway.

cAMP activates PKA which further phosphorylates CREB to regulate gene expression. In oocytes, DNA re-methylation is regulated by DNMTs including DNMT3A, DNMT3L, and DNMT1. Therefore, we next investigated whether melatonin regulates DNMTs expression via the cAMP/PKA/CREB pathway in oocytes. We examined the expression of CREB1, CREM (cAMP responsive element modulator), CREB3l2 (cAMP responsive element binding protein 3 like 2), and ATF1 (activating transcription factor 1) in oocytes. Results showed that ADCYs activator forskolin treatment significantly increased the mRNA expression of CREB1 and CREM, and that the expression of CREB3l2 and ATF1 was slightly increased (*Figure 6A*). In addition, the ADCYs inhibitor SQ22536 significantly reduced the expression of CREB1 and CREB3l2, although the expression of CREM and ATF1 was slightly decreased in oocytes (*Figure 6A*). Furthermore, treatment with the ADCYs inhibitor SQ22536 significantly reduced the concentration of pCREB1, but it was increased by forskolin treatment in oocytes (*Figure 6B and C*). The pCREB1 level was also increased by 8-Bromo-cAMP and decreased by the PKA antagonist H89 2HCL in oocytes (*Figure 6D–G*). These results suggest that the expression and phosphorylation of CREB1 can be regulated by the cAMP/PKA pathway. *Yang et al., 2021* demonstrated that CREB regulated DNMT3a expression in neurons of the dorsal root ganglion by binding to the promoter region. In the present study, the binding of pCREB1 with relative regions of DNMTs was examined using CUT & Tag assay. Each sample contained 500 GV oocytes, and two replicates were involved. The sequencing result revealed that five fragments including 10 pCREB1 binding motifs (predicted using the online tool JASPAR, *Supplementary file 4*) were associated with DNMTs, including 3 fragments at intron 1 and distill intergenic regions of DNMT3A, 1 fragment at the promoter region of DNMT1, and 1 fragment at intron 13 of DNMT3L (*Supplementary file 5*). These results suggest that pCREB1 may have contributions to regulating the expression of DNMTs.

Next, we investigated the expression of *Dnmt1*, *Dnmt3a*, and *Dnmt3l* in oocytes. The melatonin inhibitor luzindole (slightly) and the ADCYs activator forskolin (significantly) increased the expression of DNMT1 and DNMT3A, respectively (*Figure 7A and B*). Melatonin and the ADCYs inhibitor SQ22536 significantly and slightly reduced the expression of DNMT1 and DNMT3A in oocytes, respectively (*Figure 7A and B*). The protein level of DNMT3A in GV oocytes was also significantly increased by 8-Bromo-cAMP and decreased by the PKA antagonist H 89 2HCL, respectively (*Figure 7C–F*). Although DNMT1 is well-known as a maintenance methyltransferase, it also contributes to de novo methylation in oocytes (*Li et al., 2018*). Therefore, we examined the localization of DNMT1 in oocytes, and found that 8-Bromo-cAMP treatment significantly increased the localization of DNMT1 in the nucleus of oocytes, but it was reduced by the PKA antagonist H89 2HCL (*Figure 7G–J*). When the activation of DNMTs was inhibited by 5-azacytidine, the methylation level in GV oocytes was significantly

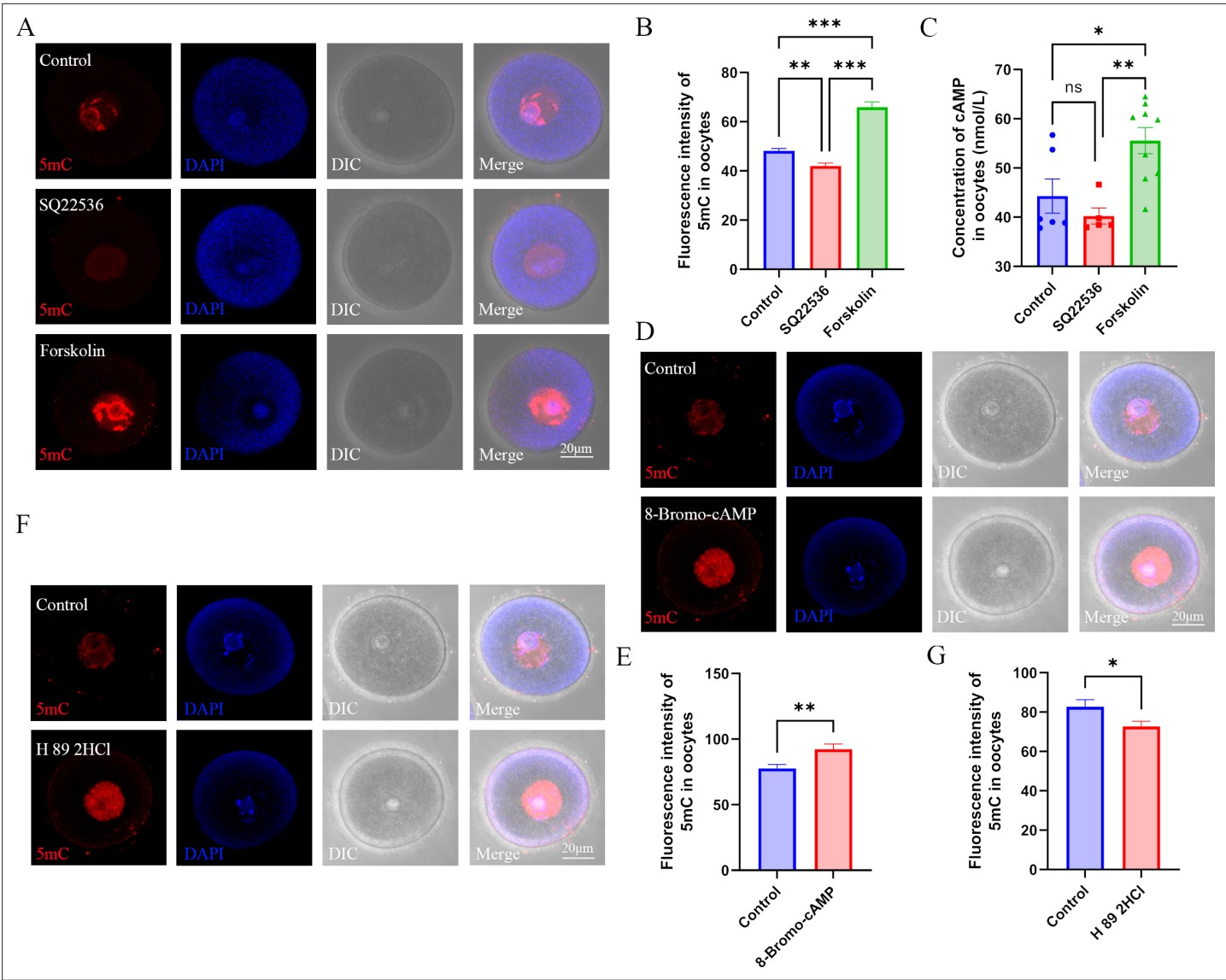

**Figure 5.** Role of cAMP in DNA methylation in oocytes. (**A**) Female mice were respectively treated with the adenylyl cyclase (ADCY) inhibitor SQ22536 or activator forskolin. Oocyte methylation was examined using immunofluorescence. (**B**) The relative intensity of fluorescence in oocytes was analyzed using Image J (Control, n=107; SQ22536, n=51, p=0.000257; Forskolin, n=57, p=3.099×10$^{-16}$). **p<0.01; ***p<0.001. Data presented as mean ± SEM; a two-tail t-test was used. (**C**) cAMP concentration in oocytes was examined using ELISA. *p<0.05; **p<0.01. Data presented as mean ± SEM; a two-tail t-test was used; n: control=6, SQ22536=5, p=0.350475, forskolin=9, p=0.001756. (**D**) Female mice were treated with the cAMP analogue 8-Bromo-cAMP, and oocyte methylation was examined using immunofluorescence. (**E**) The relative fluorescence intensity of 5mC was analyzed using Image J (Control, n=41; 8-Bromo-cAMP, n=42, p=0.004255). **p<0.01. Data presented as mean ± SEM; a two-tail t-test was used. (**F**) Female mice were treated with the PKA (protein kinase A) antagonist H 89 2HCL, and then oocyte methylation was examined using immunofluorescence. (**G**) The relative fluorescence intensity of 5mC was analyzed using Image J (Control, n=24; H 89 2HCl, n=25, p=0.032292). *p<0.05. Data presented as mean ± SEM; a two-tail t-test was used. Source data are presented in *Figure 5—source data 1*.

The online version of this article includes the following source data for figure 5:

**Source data 1.** Extended data for *Figure 5B, C, E and G*.

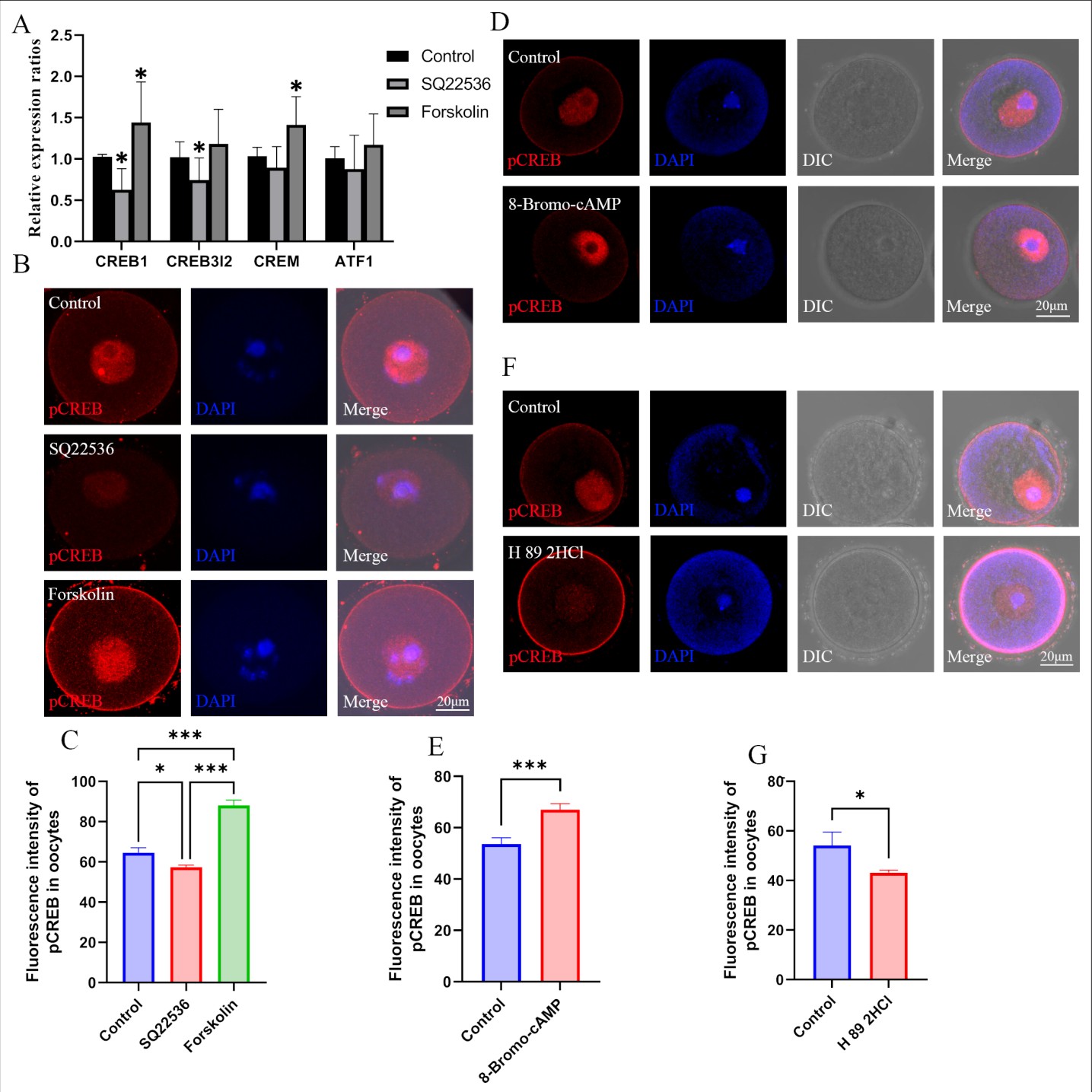

**Figure 6.** Effects of cAMP on CREB1. (**A**) The mRNA expression of cAMP-response element binding (CREB) proteins in oocytes was examined by qPCR. *p<0.05. Data presented as mean ± SD; a two-tail t-test was used. p-value presented in the *Figure 6—source data 1*. (**B**) Phosphorylated CREB1 (pCREB1) in oocytes was examined using immunofluorescence. (**C**) The relative fluorescence intensity of pCREB1 was examined by Image J (Control, n=36; SQ22536, n=48, p=0.003985; Forskolin, n=41, p=4.402×10$^{-19}$). *p<0.05; **p<0.01; ***p<0.001. Data presented as mean ± SEM; a two-tail t-test was used. (**D**) After treatment with the cAMP analogue 8-Bromo-cAMP, pCREB1 in oocytes was examined using immunofluorescence. (**E**) The relative fluorescence intensity was analyzed using Image J (Control, n=28; 8-Bromo-cAMP, n=28, p=0.00022). ***p<0.001. Data presented as mean ± SEM; a two-tail t-test was used. (**F**) Immunofluorescence of pCREB in GV oocytes after treated with H89 2HCl. (**G**) The relative intensity of pCREM in GV oocytes treated with H89 2HCl, n: control=23, H89 2HCl=26, p=0.040168. Source data are presented in *Figure 6—source data 1*.

The online version of this article includes the following source data for figure 6:

**Source data 1.** Extended data for *Figure 6A, C, E, and G*.

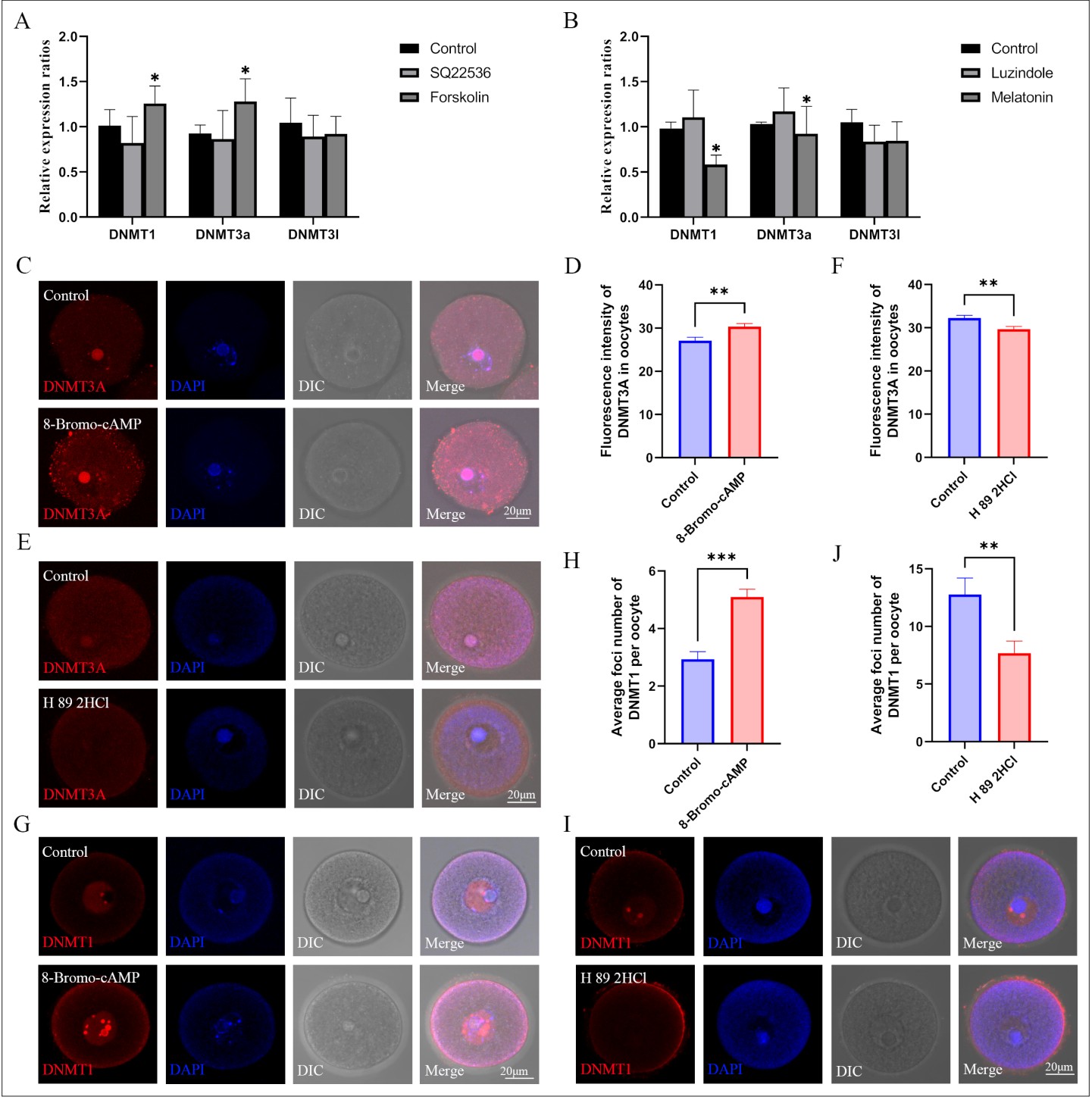

**Figure 7.** Role of the melatonin/cAMP/PKA pathway in the expression of DNMTs. (**A**) The expression levels of *Dnmt1*, *Dnmt3a*, and *Dnmt3l* in oocytes were examined using qPCR after the treatment with SQ22536 and forskolin. *p<0.05. Data presented as mean ± SEM; a two-tail t-test was used. p-value presented in the source data. (**B**) The relative expressions of *Dnmt1*, *Dnmt3a*, and *Dnmt3l* in oocytes were examined using qPCR after the treatment with luzindole and melatonin. *p<0.05. Data presented as mean ± SEM; a two-tail t-test was used. (**C**) After 8-Bromo-cAMP treatment, the relative expression of DNMT3a in oocytes was examined using immunofluorescence and calculated by Image J (**D**) (Control, n=54; 8-Bromo-cAMP, n=70, p=0.002447). **p<0.01. Data presented as mean ± SEM; a two-tail t-test was used. (**E and F**) Treatment with the protein kinase A (PKA) antagonist H 89 2HCL treatment significantly reduced the level of DNMT3A in oocytes examined using immunofluorescence (Control, n=62; H 89 2HCl, n=48, p=0.003922). **p<0.01. Data presented as mean ± SEM; a two-tail t-test was used. (**G and H**) DNMT1 localization in the oocyte nucleus was examined using immunofluorescence after 8-Bromo-cAMP treatment (Control, n=30; 8-Bromo-cAMP, n=31, p=3.136*10$^{-7}$). ***p<0.001. Data presented as mean

*Figure 7 continued on next page*

*Figure 7 continued*

± SEM;a two-tail t-test was used. (**I and J**) The localization of DNMT1 in oocyte nucleus was reduced by the treatment with the PKA antagonist H 89 2HCL (Control, n=22; H 89 2HCl, n=28, p=0.004929). ** p<0.01. Data presented as mean ± SEM; a two-tail t-test was used. Source data are presented in *Figure 7—source data 1*.

The online version of this article includes the following source data and figure supplement(s) for figure 7:

**Source data 1.** Extended data for *Figure 7A, B, D, F, H, and J*.

**Figure supplement 1.** DNMTs regulate DNA methylation in oocytes.

decreased (*Figure 7—figure supplement 1*). These results suggest that melatonin may influence the genomic methylation of oocytes by regulating the expression of DNMT1 and DNMT3A by the cAMP/PKA/CREB pathway.

## Increased DNMTs mediate hyper-methylation of HFD oocytes via cAMP/PKA/CREB pathway

To explore how melatonin regulates the hyper-methylation of HFD oocytes, we examined the expression of DNMTs. Results showed that maternal obesity in mice significantly increased the expression of DNMT1, DNMT3A, and DNMT3L in oocytes (*Figure 8A*). Thus, we aimed to confirm whether the cAMP/PKA/CREB pathway mediated the increased expression of DNMTs in HFD oocytes. We examined the concentration of cAMP in oocytes, and found that maternal obesity significantly increased the concentration of cAMP in oocytes compared with that in the CD group (*Figure 8B*). The mRNA expression of CREB1, but not CREM, in the HFD oocytes was significantly increased compared with that in the CD group (*Figure 8C*), and the pCREB1 level in HFD oocytes was also significantly increased (*Figure 8D and E*). However, the increased level of pCREB1 was reduced by exogenous melatonin treatment (*Figure 8D and E*). When obese females were treated with the PKA antagonist H89 2HCL, both 5mC and pCREB1 levels were significantly reduced in oocytes (*Figure 8F–I*). These results suggest that reduced melatonin in obese mice may increase the expression of DNMTs via the cAMP/PKA/CREB pathway.

The increased expression of DNMT3a (*Figure 8A*) may have contributed to the genomic hyper-methylation of HFD oocytes because the primary function of DNMT3A is de novo DNA methylation. The de novo methylation function of DNMT1 is usually prevented by *Stella* (also known as *Dppa3* or *Pgc7*) in oocytes (*Bostick et al., 2007*). Nevertheless, maternal obesity significantly reduces the expression of *Stella* in oocytes (*Han et al., 2018*), which indicates that DNMT1 may also contribute to the hyper-methylation of the HFD oocytes. We found that the localization of DNMT1 in the nuclei of HFD oocyte was significantly increased, which could be reduced by the PKA antagonist H89 2HCL treatment (*Figure 8J and K*). These results suggest that the decreased melatonin induced by maternal obesity increases the expression of DNMTs via the cAMP/PKA/CREB pathway, which results in the hyper-methylation in HFD oocytes.

## Discussion

Maternal obesity has negative effects on oocyte quality and offspring health, but the underlying mechanisms are not well understood. In the present study, we found that maternal obesity-induced hyper-methylation in oocytes, and that the abnormal methylation, at least in part, is transmitted to F2 oocytes through females, which may be associated with the occurrence and inheritance of the metabolic disorders. Maternal obesity-induced metabolic changes in the serum. The decreased melatonin in serum may be involved in regulating the hyper-methylation of HFD oocytes by increasing the expression of DNMTs, which is mediated by the cAMP/PKA/CREB pathway.

Transgenerational epigenetic inheritance is common in plants, but related investigations in mammals are hindered by epigenetic reprogramming events during gametogenesis and early embryo development (*Schmitz and Ecker, 2012*; *Xavier et al., 2019*). In mammals, isogenic agouti viable yellow ($A^{vy}$) and axin-fused ($Axin^{Fu}$) mice, whose phenotypes are regulated by the DNA methylation level of IAP (intra-cisternal A particle long terminal repeat) respectively located at the upstream and in intron 6, are solid evidence confirming the transgenerational inheritance of epigenetic modifications (*Morgan et al., 1999*; *Rakyan et al., 2002*). Nevertheless, epigenetic modifications can be

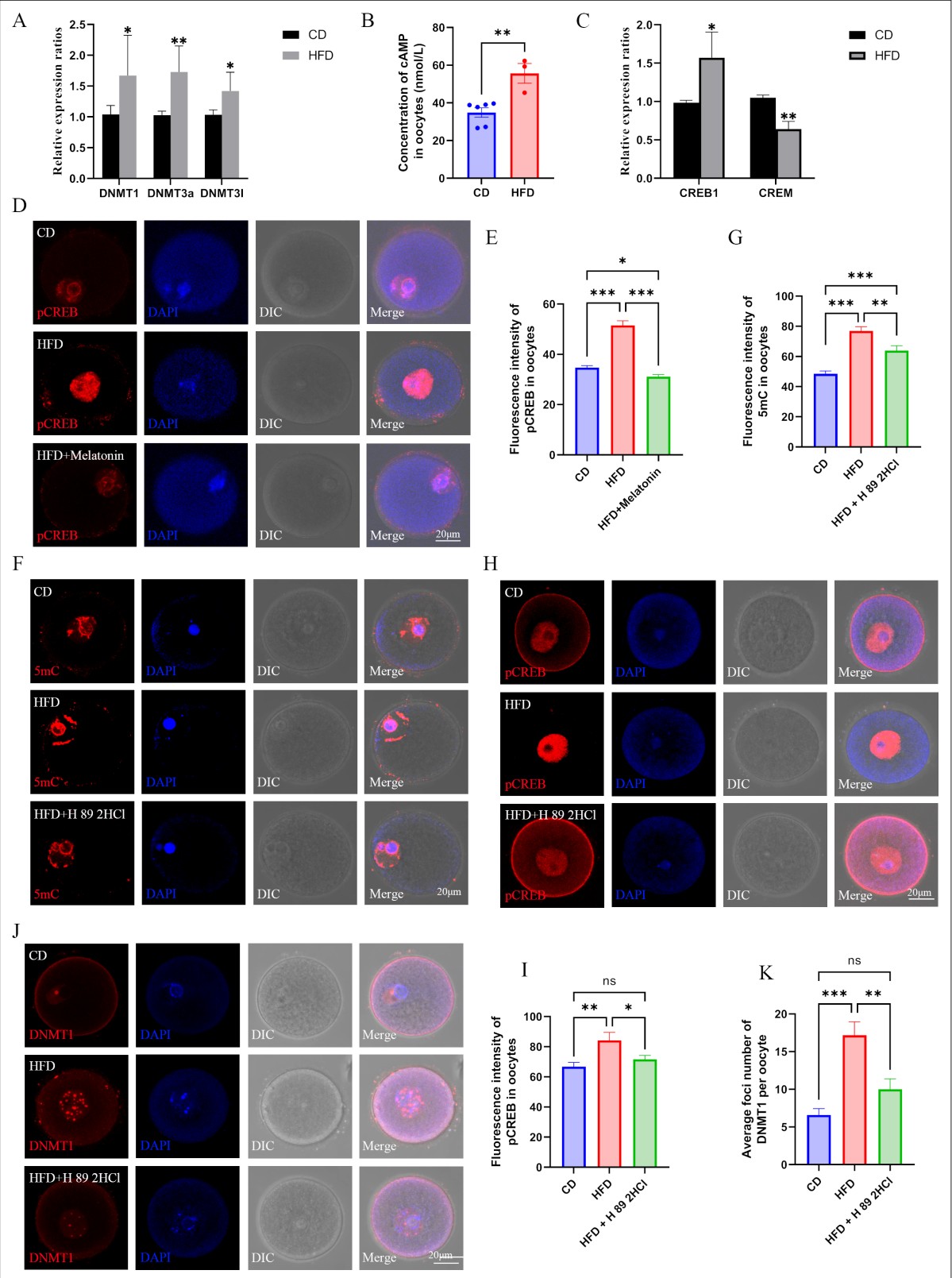

**Figure 8.** Melatonin regulates DNMTs expression via cAMP/PKA/CREB pathway in high-fat diet (HFD) oocytes. (**A**) The relative expression of *Dnmt1*, *Dnmt3a,* and *Dnmt3l* in HFD oocytes was examined using qPCR. *p<0.05; **p<0.01. Data presented as mean ± SD; two-tail t-test used. p-value presented in the source data. (**B**) The concentration of cAMP in HFD oocytes was examined using ELISA. **p<0.01. Data presented as mean ± SEM; a two-tail t-test was used, n: CD=6, HFD = 3, p=0.004375.(**C**) The relative expressions CREB1 and CREM in HFD oocytes were tested using qPCR. *p<0.05;

*Figure 8 continued on next page*

*Figure 8 continued*

**p<0.01. Data presented as mean ± SD; a two-tail t-test was used. Replicated three times for each gene, and p value presented in the source data. (**D and E**) The level of pCREB1 in oocytes was examined using immunofluorescence, and the relative fluorescence intensity was calculated by Image J (**E**) (CD, n=69; HFD, n=49, p=3.326×10$^{-16}$; HFD + melatonin, n=61, p=8.997×10$^{-20}$). HFD, oocytes from obese mice; CD, oocytes from control mice; HFD + melatonin, oocytes from obese mice treated with exogenous melatonin. *p<0.05; ***p<0.001. Data presented as mean ± SEM; a two-tail t-test was used. (**F and G**) Treatment with the PKA antagonist H89 2HCL reduced the methylation level of HFD oocytes (CD, n=48; HFD, n=31, p=1.674*10$^{-13}$; HFD + H 89 2HCl, n=27, p=0.00324). ** p<0.01; *** p<0.001. Data presented as mean ± SEM; a two-tail t-test was used. (**H and I**) The level of pCREB1 in HFD oocytes was also decreased by the treatment with the protein kinase A (PKA) antagonist H89 2HCL (CD, n=17; HFD, n=17, p=0.006249; HFD +H 89 2HCl, n=22, p=0.027987). *p<0.05; **p<0.01; ns, no statistical significance between groups. Data presented as mean ± SEM; a two-tail t-test was used. (**J and K**) Treatment with the PKA antagonist H89 2HCL reduced the localization of DNMT1 in HFD oocytes (CD, n=24; HFD, n=29, p=6.214×10$^{-6}$; HFD + H 89 2HCl, n=25, p0.003147). **p<0.01; ***p<0.001; ns, no statistical significance between groups. Data presented as mean ± SEM; a two-tail t-test was used. Source data are presented in *Figure 8—source data 1* .

The online version of this article includes the following source data for figure 8:

**Source data 1.** Extended data for *Figure 8A, B, C, E, G, I, and K*.

affected by environmental factors such as metabolic diseases and diet. The intergenerational inheritance of phenotypes and epigenetic changes induced by maternal environmental factors have been confirmed by previous studies (*Chen et al., 2022a*; *Ge et al., 2014*), but there are still many debates about the transgenerational inheritance of epigenetic changes induced by the environment. Rats from stressed mothers are more likely to be stressed, and that can be transmitted across generations, but this transgenerational inheritance is not mediated by gametes (*Weaver et al., 2004*). Females cannot mediate the transgenerational inheritance of hyper-methylation induced by diet in $A^{vy}$ mice (*Waterland et al., 2007*), but another study reported that if the $A^{vy}$ allele is from the father, the hyper-methylation induced by diet during pregnancy can be retained in germ cells (*Cropley et al., 2006*). *Anway et al., 2005* reported that the abnormal spermatogenesis induced by the exposure to vinclozolin during pregnancy can be transmitted across four generations via sperm. We previously reported that maternal obesity disturbed the DNA methylation status of imprinted genes in oocytes (*Ge et al., 2014*), but the transgenerational inheritance was not observed. We also demonstrated that disturbed methylation in oocytes induced by undernourishment in utero could be inherited, at least partly, by F2 oocytes via females (*Tang et al., 2023*). Recently, *Takahashi et al., 2023* edited DNA methylation of promoter-associated CGIs, and found that the edited DNA methylation, associated with disrupted metabolism, was stably inherited by multiple generations. In the present study, we found that maternal obesity-induced genomic hyper-methylation in oocytes, and that a part of the abnormal methylation was transmitted to F2 via female gametes. Moreover, the transmission of metabolic disorders has also been observed across two generations. These results suggest that the transgenerational inheritance of abnormal methylation induced by maternal obesity, at least in part, can be mediated by oocytes, which may be a reason for the inheritance of metabolic disorders.

During the methylation process, the methyl group is donated by SAM which is generated from homocysteine, 5-methyltetrahydrofolate (5mTHF), and methionine. 5mTHF is an intermediate of one-carbon metabolism (*Mentch and Locasale, 2016*). One-carbon units such as folate and vitamin B12 are crucial for the establishment of methylation (*Mentch and Locasale, 2016*). Disturbed glucose and lipid metabolism also have a negative influence on DNA methylation (*Keller et al., 2014*). These indicate that abnormal metabolism induced by maternal obesity (*King, 2006*) may play a key role in the genomic hyper-methylation in oocytes. In the present study, we found that the metabolomics of serum from HFD mice was distinguishable from that of serum from CD mice. Although the concentrations of vitamin B6 and methionine were higher in HFD serum than that in CD, this may not be an important reason for the genomic hyper-methylation in oocytes because the concentrations of SAM, SAH, and HCY in the livers and oocytes were similar between CD and HFD mice. In humans, obesity reduces the melatonin level in circulation (*Overberg et al., 2022*; *Virto et al., 2018*). In the present study, we also found that maternal obesity induced by a high-fat diet reduced the concentration of melatonin in the serum. Melatonin not only can decrease body weight, but also regulates DNA methylation of somatic cells and germ cells (*Davoodvandi et al., 2022*; *Lan et al., 2018*). Nevertheless, the molecular mechanism by which melatonin regulates DNA methylation in oocytes is still unclear. Melatonin has two receptors, MT1 and MT2, both of which have been identified in oocytes (*Wang et al., 2021*). Melatonin receptors interact with the inhibitor G-protein and can regulate gene expression via the

cAMP/PKA/CREB pathway (*Wongprayoon and Govitrapong, 2021*). CREM and cAMP mediate DNA methylation in somatic cells by regulating the expression of *Dnmt3a* (*Fang et al., 2015*; *Hedrich et al., 2014*). In neurons, CREB interacts with the promoter of *Dnmt3a* to regulate its expression and DNA methylation (*Yang et al., 2021*). In the present study, we found that melatonin regulates methylation in oocytes by increasing the expression of DNMT1 and DNMT3a, which is mediated by the cAMP/PKA/CREB pathway. Similar results were also observed in HFD oocytes. Hyper-methylation of HFD oocytes can be reduced by exogenous melatonin and PKA inhibitors. These suggest that decreased melatonin levels are involved in regulating the genomic hyper-methylation of HFD oocytes by increasing the expression of DNMTs, which is mediated by the cAMP/PKA/CREB pathway.

During follicular development, re-methylation in oocytes is catalyzed mainly by DNMT3A and DNMT3L (*Kibe et al., 2021*). DNMT1 is usually responsible for maintaining DNA methylation, but DNMT1 also contributes to CG methylation in oocytes (*Shirane et al., 2013*). During normal oocyte development, DNMT1 is mainly prevented from the nuclei by *Stella*. When the *Stella* level is knocked out, *Uhrf1* (Ubiquitin-like containing PHD Ring Finger 1) moves to the nucleus from the cytoplasm and recruits DNMT1 to chromatin, resulting in hyper-methylation (*Li et al., 2018*). Maternal obesity significantly decreases the expression of *Stella* in oocytes (*Han et al., 2018*). These suggest that maternal obesity may induce hyper-methylation in oocytes. In the present study, we found that maternal obesity increased the genomic DNA methylation in GV and MII oocytes, the expression of DNMT1, and its localization at chromatin in GV oocytes. These suggest that reduced *Stella* in HFD oocytes may recruit more DNMT1 into chromatin resulting in hyper-methylation. Nevertheless, Han LS et al. reported that maternal obesity has no significant influence on whole genome methylation of GV oocytes and Hou YJ et al. reported that maternal obesity reduces whole genome methylation of NSN (no Hoechst-positive rim surrounding the nucleolus) GV oocytes. This contradiction may be associated with the methods used to examine the methylation level and the sample size.

In summary, we found that maternal obesity-induced genomic hyper-methylation in oocytes and that at least a part of the altered methylation can be transmitted to F2 oocytes, which may be a reason for the inheritance of metabolic disorders. Furthermore, reduced melatonin in HFD mice was involved in regulating the genomic hyper-methylation of oocytes by increasing the expression of DNMTs, and this process was mediated by the cAMP/PKA/CREB pathway. However, there are some limitations for the present study: there is not enough evidence to confirm the role of altered DNA methylation in metabolic disorders in the offspring of obese mothers The molecular mechanisms by which DNA methylation escapes reprogramming in oogenesis have not been elucidated. There are may be other mechanisms involved in regulating genomic hyper-methylation in HFD oocytes. Therefore, more studies are needed in the short future.

## Materials and methods
### Mice
C57BL/6 mice were purchased from Jinan Pengyue Company (Jinan, China). Mice were housed in the controlled room with 12 hr light and 12 hr dark cycle, and at 23–25°C. The Animal Ethics Committee of Qingdao Agricultural University supported all procedures (QAU201900326).

For the obesity model, female C57BL/6 mice at the age of 4 wk were randomly divided into two groups fed with a high-fat diet (HFD, Research Diets, D12492, USA) and normal diet (CD) for 12 wk, respectively. We examined the body weight every week. The formulation of the diets is presented in *Supplementary file 6*.

Offspring were produced according to the schedule in *Figure 1J*. For F1 offspring, female HFD and CD mice were mated with normal adult male C57BL/6 mice, respectively, and the offspring were marked as HF1 and CF1. To avoid the effects of males on methylation, the same males were used to produce F2: female HF1 and CF1 mating with normal males, marked as HF2 and CF2.

### Immunofluorescence
Briefly, oocytes were fixed with 4% PFA (paraformaldehyde), permeabilized with 0.5% TritonX-100, and blocked with 1% BSA (bovine serum albumin). After that, the oocytes were incubated with primary antibodies overnight at 4°C. The secondary antibodies were stained for 1 hr at room temperature. For 5mC and 5hmC staining, the oocytes were treated with 4 N HCl for 10 min after permeabilization, and

then transferred into 100 mM Tris-HCl for 10 min. After washed three times using PBS/BSA with 0.05% Tween 20, the oocytes were blocked with PBS/BSA for 1 hr, and incubated with primary antibodies overnight at 4°C. Secondary antibodies were stained for 1 hr at room temperature. Then, oocytes were transferred into DAPI with mounting and sealed. Fluorescence signal was examined using a laser scanning confocal microscope (Leica SP5, Germany). The relative fluorescence intensity was examined using Image J.

## Antibodies

Primary antibodies used in the present study included anti-5mC antibody (Abcam, ab73938, USA), anti-5hmC antibody (Abcam, ab214728, USA), anti-pCREB antibody (Cell Signaling Technology, 9198 S, USA), anti-DNMT3a antibody (Active motif, 61478, Shanghai, China), and anti-DNMT1 antibody (Active motif, 39204, Shanghai, China).

## Whole-genome bisulfite sequencing and analysis

Metaphase II (MII) oocytes were collected from the oviduct. For each sample, 100 oocytes from at least 10 mice were pooled together and transferred to a lysis buffer. Genomic DNA was fragmented, and the end was repaired. Then, fragmentations were ligated with adapters. Bisulfite treatment was performed using EZ DNA Methylation-Direct (Zymo Research, USA). Lambda DNA was used as a control. After that, the sequencing library was established and sequenced using Illumina HiSeq/NovaSeq (Novogene, Shanghai, China). The raw data quality was evaluated using FastQC, and low-quality data and adapters were trimmed using fastp. Clean data were compared to the reference genome mm10. Methylated C site calling was performed using Bismark. Differentially methylated regions (DMRs) were identified using DSS-single. The enrichment of genes in the KEGG pathway was carried out using the online tool KOBAS.

## qPCR

Total RNA was extracted from oocytes or tissues using RNAprep Pure Micro Kit (Tiangen, DP420, Beijing, China) or RNA Easy Fast Kit (Tiangen, DP451, Beijing, China). cDNA was synthesized using the Hifair III 1st Strand cDNA Synthesis Kit (Yeasen, Shanghai, China). cDNA was used as a templates to examine the relative expression of genes. Housekeeping genes *Ppia* and *Gapdh* were used as references. Relative expression was calculated as $2^{-\Delta\Delta Ct}$.

## BS

Each sample included five oocytes and at least 20 samples were used for each DMR. The samples were treated as described in a previous study (*Ge et al., 2014*). Briefly, samples were treated with lysis buffer and 0.3 M NaOH, respectively. After that, samples were embedded in 2% low melting point agarose (Sigma), which was treated with fresh bisulfite solution (2.5 M sodium metabisulfite, Merck; 125 mM hydroquinone, Sigma; pH 5) for 4 hr. Treated DNA was used as templates to amplify the target fragment using nest-PCR. PCR products were cloned to T-vector and sequenced. Methylation status was analyzed using BiqAnalyzer, which can remove the low conversion rate (<97%) and possible repeat sequences. At least 10 available clones were used for each DMR.

## Glucose and insulin tolerance

Glucose and insulin tolerance (GTT and ITT) were examined as previously reported (*Tang et al., 2023*). Briefly, mice were treated with glucose at 2 g/kg body weight or insulin (Actrapid, Novo Nordisk, Denmark) at 10 IU/kg body weight after 16 hr or 4 hr fasting, respectively. After that, blood glucose was measured by tail blood at 0, 30, 60, 90, and 120 min, respectively.

## ELISA

Concentrations of cAMP, SAM, SAH, and HCY were examined using ELISA kits (Jinma Biotechnology Co. Ltd, Shanghai, China) according to the handbook. A standard curve was produced using four-parameter logistics.

## Non-target metabolomics in serum

Metabolites in the serum were examined using LC-MS/MS (BGI, Wuhan, China). Raw date was treated with Compound Discover 3.1 (Thermo Fisher Scientific, USA). After that, preprocessing of the

exported data was performed using metaX. Metabolites were identified according to the databases of BMDB (BGI), mzCloud, and ChemSpider (HMDB, KEGG, LipidMaps). Identified metabolites were annotated according to KEGG and HMDB. Differential metabolites were scanned using PCA and PLS-DA combined with fold changes and Student's t-test.

### Chemicals

Inhibitors used in the present study included luzindole (Sigma, Shanghai, China), SQ22536 (Selleck, Shanghai, China), forskolin (Selleck, Shanghai, China), H89 2HCL (Selleck, Shanghai, China), and azacitidine (Selleck, Shanghai, China). 8-Bromo-cAMP was purchased from Selleck. Melatonin (Sigma) was injected by the tail vein for 14 d. The other chemicals were administrated via intraperitoneal injection for 14 d. The control group was injected with the appropriate solutions.

### CUT & Tag and sequencing

For each sample, 500 oocytes were pooled together for this experiment, and two replicates were performed. Library was established using Hyperactive Universal CUT&Tag Assay Kit (Vazyme, Shanghai, China) according to the manufacturer's instructions. Briefly, oocytes were washed with washing buffer and transferred into tubes with ConA Beads Pro for 10 min at room temperature. The tubes were placed on a magnetic rack for 2 min, and then discarded the supernatant. After that, 50 µl of precooled antibody buffer and primary antibody were added and incubated overnight at 4 °C. Then, tubes were put on a magnetic rack for 2 min, discarded supernatant, and added 50 µl Dig-wash buffer with secondary antibody (1:100) into tubes incubated at room temperature for 60 min. Samples were washed three times, and then incubated with pA/G-Tnp pro for 1 h at room temperature. After washing, 50 µl of TTBL was added to the samples and incubated at 37°C for 1 hr. Then, 2 µl 10% SDS and DNA Spike-in were added to the samples, which were incubated at 55°C for 10 min. Supernatant was transferred into a new tube after being put on a magnetic rack for 2 min. 25 µl DNA Extract Beads Pro was added into the supernatant, incubated for 20 min at room temperature, and then washed two times with B&W buffer. After that, the DNA Extract Beads Pro was re-suspended in 15 µl ddH$_2$O, and amplified at 60°C for 20 cycles. Library quality was examined using Qubit, AATI, and QPCR, and then sequenced using NovaSeq 6000 (Novogene, Shanghai, China). Adaptor and low-quality reads were removed from the raw data, and clean data were used for further analysis. Reads were mapped to the mouse reference genome mm39 using Bowtie2. Peak calling was performed using MACS2 at q-value <0.05. The peak was subsequently annotated into related gene regions.

### Statistical analysis

Average data are presented as mean ± SEM (standard error), and the statistical difference was calculated using two-tail independent-samples t-test. Methylation level is presented as a percentage, and the statistical difference was calculated using a the chi-square test. If p-value <0.05, the statistical difference was considered to be significant. Repeated at least three times for each experiment, and each assay has at least three biological replicates.

## Acknowledgements

This work was supported by the National Natural Science Foundation of China (31872312), the National R&D Program of China (2022YFC2703500), the Breeding Plan of Shandong Provincial Qingchuang Research Team (Innovation Team of Farm Animal Cloning 012–1622001), and the Doctor Foundation of Qingdao Agricultural University (6631116008). We acknowledge the Central Laboratory Life Science Instrument Platform of the College of Life Science and the Central Platform of Qingdao Agricultural University for provding the fluorescence microscope and the instrument for qPCR, and thank Tao Xu and Lin Song for their invaluable help in train the students how to use these instruments.

## Additional information

### Funding

| Funder | Grant reference number | Author |
|---|---|---|
| National Natural Science Foundation of China | 31872312 | Zhao-Jia Ge |
| National R&D Program of China | 2022YFC2703500 | Qing-Yuan Sun |
| Breeding Plan of Shandong Provincial Qingchuang Research Team | 012-1622001 | Ming-Hui Zhao |
| Doctor Foundation of Qingdao Agricultural University | 6631116008 | Zhao-Jia Ge |

The funders had no role in study design, data collection and interpretation, or the decision to submit the work for publication.

### Author contributions

Shuo Chao, Conceptualization, Formal analysis, Validation, Investigation, Methodology; Jun Lu, Conceptualization, Data curation, Formal analysis, Validation, Investigation; Li-Jun Li, Data curation, Formal analysis, Validation; Hong-Yan Guo, Formal analysis, Methodology; Kuipeng Xu, Formal analysis; Ning Wang, Xiao-Wen Jin, Shao-Ge Wang, Methodology; Shu-Xian Zhao, Validation, Methodology; Shen Yin, Conceptualization, Supervision, Writing – review and editing; Wei Shen, Gui-An Huang, Supervision; Ming-Hui Zhao, Supervision, Funding acquisition; Qing-Yuan Sun, Supervision, Funding acquisition, Project administration, Writing – review and editing; Zhao-Jia Ge, Conceptualization, Data curation, Formal analysis, Supervision, Funding acquisition, Writing - original draft, Project administration, Writing – review and editing

### Author ORCIDs

Shuo Chao ⓘ https://orcid.org/0009-0009-2456-6926
Qing-Yuan Sun ⓘ https://orcid.org/0000-0002-0148-2414
Zhao-Jia Ge ⓘ https://orcid.org/0000-0002-6714-6675

### Ethics

All of the animals were handled according to approved institutional animal care and use committee protocols of the Qingdao Agricultural University. The Animal Ethics Committee of Qingdao Agricultural University supported all procedures (QAU201900326). Mice were sacrificed by cervical dislocation, and every effect was made to minimize suffering.

Joint Public review: https://doi.org/10.7554/eLife.97507.3.sa1
Author response https://doi.org/10.7554/eLife.97507.3.sa2

## Additional files

### Supplementary files

- Supplementary file 1. Some information on whole genome bisulfite sequencing.
- Supplementary file 2. Kyoto Encyclopedia of Genes and Genomes (KEGG) pathway analysis of metabolism-relative genes.
- Supplementary file 3. Differentially methylated region (DMRs) methylation status of metabolism-relative genes.
- Supplementary file 4. Binding sites of CREB1 on sequences of DNMTs.
- Supplementary file 5. Annotation of peaks at relative gene regions.
- Supplementary file 6. Percentage of ingredients of diets.

• MDAR checklist

## Data availability

Sequencing data have been deposited in NGDC Genome Sequence Archive under accession codes CRA011654.

The following dataset was generated:

| Author(s) | Year | Dataset title | Dataset URL | Database and Identifier |
|---|---|---|---|---|
| Chao S, Lu J, Guo HY, Li LJ, Xu KP, Wang N, Zhao SX, Jin XW, Wang SG, Yin S, Shen W, Zhao MH, Huang HA, Sun QY, Ge ZJ | 2024 | Inheritance and mechanism of disturbed DNA methylation in oocytes induced by maternal obesity | https://bigd.big. ac.cn/gsa/browse/ CRA011654 | Genome Sequence Archive, CRA011654 |

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

# Appendix 1

## Appendix 1—key resources table

| Reagent type (species) or resource | Designation | Source or reference | Identifiers | Additional information |
|---|---|---|---|---|
| Biological sample (*Mus musculus*) | *Mus musculus* | C57BL/6 | C57BL/6 | Jinan Pengyue Company (Jinan, China) |
| Antibody | Anti-5mC antibody (Mouse monoclonal) | Abcam | Ab73938 | IF(1:200) |
| Antibody | Anti-5hmC antibody (Rabbit monoclonal) | Abcam | Ab214728 | IF(1:200) |
| Antibody | Anti-DNMT3A antibody (Rat Monoclonal) | Active motif | 61478 | IF(1:200) |
| Antibody | Anti-DNMT1 antibody (Mouse Monoclonal) | Active motif | 39204 | IF(1:200) |
| Antibody | Anti-pCREB antibody (Rabbit mAb) | Cell Signaling Technology | 9198 S | IF(1:400) |
| Sequence-based reagent | Bhlha15_F1 | This paper | PCR primers | GTAGGGTGGTTTATTTTAGATT |
| Sequence-based reagent | Bhlha15_R1 | This paper | PCR primers | ACCATCCCATCTATCTATCTAT |
| Sequence-based reagent | Bhlha15_F2 | This paper | PCR primers | TTTGGTAAGTTTTTAGAGAGGT |
| Sequence-based reagent | Bhlha15_R2 | This paper | PCR primers | CCCAACAATCCTATATAATTTC |
| Sequence-based reagent | Mgat1_F1 | This paper | PCR primers | TAGTAGAGGAAGGTTTTGGA |
| Sequence-based reagent | Mgat1_R1 | This paper | PCR primers | CCTTATCCTCCTAAAACAAAC |
| Sequence-based reagent | Mgat1_F2 | This paper | PCR primers | GGAAGGTTTTGGAAGGAG |
| Sequence-based reagent | Mgat1_R2 | This paper | PCR primers | ACAAACCCCAAAACTAAAAAC |
| Sequence-based reagent | Pik3cd_F1 | This paper | PCR primers | GTTAGAGGAGATATAGGGATTT |
| Sequence-based reagent | Pik3cd_R1 | This paper | PCR primers | CCTTAACCCCTAACTAAAATAT |
| Sequence-based reagent | Pik3cd_F2 | This paper | PCR primers | AGAGGAGATATAGGGATTTTA |
| Sequence-based reagent | Pik3cd_R2 | This paper | PCR primers | TTAACCCCTAACTAAAATATATCT |
| Sequence-based reagent | Tkt_F1 | This paper | PCR primers | TATTTTGTTGTTATTGTTTGTG |
| Sequence-based reagent | Tkt_R1 | This paper | PCR primers | CTTCAAAACCTAAAACTTCTACT |
| Sequence-based reagent | Tkt_F2 | This paper | PCR primers | TATTTTGTTGTTATTGTTTGTGT |
| Sequence-based reagent | Tkt_R2 | This paper | PCR primers | TAAAAACAAAAACACAAAACC |
| Sequence-based reagent | Pld1_F1 | This paper | PCR primers | AGGATATTTGGATAGAAGAAAG |
| Sequence-based reagent | Pld1_R1 | This paper | PCR primers | CAAAAAAACTTCAAAAACAA |
| Sequence-based reagent | Pld1_F2 | This paper | PCR primers | AGGATATTTGGATAGAAGAAAG |
| Sequence-based reagent | Pld1_R2 | This paper | PCR primers | TCCTACRAACTCAAAAATC |
| Sequence-based reagent | H19_F1 | This paper | PCR primers | GAGTATTTAGGAGGTATAAGAATT |
| Sequence-based reagent | H19_R1 | This paper | PCR primers | ATCAAAAACTAACATAAACCCCT |
| Sequence-based reagent | H19_F2 | This paper | PCR primers | GTAAGGAGATTATGTTTATTTTTGG |
| Sequence-based reagent | H19_R2 | This paper | PCR primers | CCTCATTAATCCCATAACTAT |
| Sequence-based reagent | Gapdh_F | This paper | PCR primers | CCTTCCGTGTTCCTACCC |
| Sequence-based reagent | Gapdh_R | This paper | PCR primers | CAACCTGGTCCTCAGTGTAG |
| Sequence-based reagent | Pld1_F | This paper | PCR primers | TCGTTTTGTGGACTGAGAACAC |

*Appendix 1 Continued on next page*

*Appendix 1 Continued*

| Reagent type (species) or resource | Designation | Source or reference | Identifiers | Additional information |
|---|---|---|---|---|
| Sequence-based reagent | Pld1_R | This paper | PCR primers | GCTGCTGTTGAAACCCAAATC |
| Sequence-based reagent | Bhlha15_F | This paper | PCR primers | GCTGACCGCCACCATACTTAC |
| Sequence-based reagent | Bhlha15_R | This paper | PCR primers | TGTGTAGAGTAGCGTTGCAGG |
| Sequence-based reagent | Dgka_F | This paper | PCR primers | GATGAACAGATTTTGCCAGGGA |
| Sequence-based reagent | Dgka_R | This paper | PCR primers | GTAGCAGTACACATCACTGAGAC |
| Sequence-based reagent | Pdpk1_F | This paper | PCR primers | GTGCCCATTCAGTCCAGTGT |
| Sequence-based reagent | Pdpk1_R | This paper | PCR primers | AAGGGGTTGGTGCTTGGTC |
| Sequence-based reagent | Mgat1_F | This paper | PCR primers | TTGTGCTTTGGGGTGCTATCA |
| Sequence-based reagent | Mgat1_R | This paper | PCR primers | CCACAGTGGGAACTCTCCA |
| Sequence-based reagent | Taok3_F | This paper | PCR primers | TTGCATGAAATTGGACATGGGA |
| Sequence-based reagent | Taok3_R | This paper | PCR primers | CGATGGTGTTAGGATGCTTCAG |
| Sequence-based reagent | Igf1_F | This paper | PCR primers | CTGGACCAGAGACCCTTTGC |
| Sequence-based reagent | Igf1_R | This paper | PCR primers | GGACGGGGACTTCTGAGTCTT |
| Sequence-based reagent | Map3k8_F | This paper | PCR primers | ATGGAGTACATGAGCACTGGA |
| Sequence-based reagent | Map3k8_R | This paper | PCR primers | GGCTCTTCACTTGCATAAAGGTT |
| Sequence-based reagent | Pld1_F | This paper | PCR primers | TCGTTTTGTGGACTGAGAACAC |
| Sequence-based reagent | Pld1_R | This paper | PCR primers | GCTGCTGTTGAAACCCAAATC |
| Sequence-based reagent | Tkt_F | This paper | PCR primers | ATGGAAGGTTACCATAAGCCAGA |
| Sequence-based reagent | Tkt_R | This paper | PCR primers | TGCAGCATGATGTGGGGTG |
| Sequence-based reagent | Pik3cd_F | This paper | PCR primers | GTAAACGACTTCCGCACTAAGA |
| Sequence-based reagent | Pik3cd_R | This paper | PCR primers | GCTGACACGCAATAAGCCG |
| Sequence-based reagent | Sphk2_F | This paper | PCR primers | CACGGCGAGTTTGGTTCCTA |
| Sequence-based reagent | Sphk2_R | This paper | PCR primers | CTTCTGGCTTTGGGCGTAGT |
| Sequence-based reagent | PPIA_F | This paper | PCR primers | GCCATCACCATCTTCCAGG |
| Sequence-based reagent | PPIA_R | This paper | PCR primers | CACGCCCATCACAAACAT |
| Sequence-based reagent | ADCY5_F | This paper | PCR primers | CTTGGGGAGAAGCCGATTCC |
| Sequence-based reagent | ADCY5_R | This paper | PCR primers | ACCGCTTAGTGGAGGGTCT |
| Sequence-based reagent | ADCY6_F | This paper | PCR primers | TGAGTCTTCTAGCCAGCTCTG |
| Sequence-based reagent | ADCY6_R | This paper | PCR primers | CAGCACCAAGTAGGTGAACCC |
| sequence-based reagent | ADCY9_F | This paper | PCR primers | CAACAGCGTGAGGGTCAAGAT |
| Sequence-based reagent | ADCY9_R | This paper | PCR primers | CATGGAGTCGAATTTGGGGTC |
| Sequence-based reagent | CREB1_F | This paper | PCR primers | AGCAGCTCATGCAACATCATC |
| Sequence-based reagent | CREB1_R | This paper | PCR primers | AGTCCTTACAGGAAGACTGAACT |
| Sequence-based reagent | CREB3l2_F | This paper | PCR primers | CATGTACCACACGCACTTCTC |
| Sequence-based reagent | CREB3l2_R | This paper | PCR primers | CCACCTCCATTGACTCGCTC |
| Sequence-based reagent | CREM_F | This paper | PCR primers | TTGCCCCAAGTCACATGGC |
| Sequence-based reagent | CREM_R | This paper | PCR primers | ACTGCGACTCGACTCTCAAGA |
| Sequence-based reagent | ATF1_F | This paper | PCR primers | GATTCCCACAAGAGTAACACGAC |

*Appendix 1 Continued on next page*

*Appendix 1 Continued*

| Reagent type (species) or resource | Designation | Source or reference | Identifiers | Additional information |
|---|---|---|---|---|
| Sequence-based reagent | ATF1_R | This paper | PCR primers | CCTATGCTGTCAGATGAGTCCT |
| Sequence-based reagent | DNMT1_F | This paper | PCR primers | ATCCTGTGAAAGAGAACCCTGT |
| Sequence-based reagent | DNMT1_R | This paper | PCR primers | CCGATGCGATAGGGCTCTG |
| Sequence-based reagent | DNMT3a_F | This paper | PCR primers | GAGGGAACTGAGACCCCAC |
| Sequence-based reagent | DNMT3a_R | This paper | PCR primers | CTGGAAGGTGAGTCTTGGCA |
| Sequence-based reagent | DNMT3l_F | This paper | PCR primers | GCTCTAAGACCCTTGAAACCTTG |
| Sequence-based reagent | DNMT3l_R | This paper | PCR primers | GTCGGTTCACTTTGACTTCGTA |
| Commercial assay or kit | cAMP ELISA kit | Jinma Biotechnology Co. Ltd, Shanghai, China | JS772-Mu | |
| Commercial assay or kit | SAM ELISA kit | Jinma Biotechnology Co. Ltd, Shanghai, China | JS1361-Mu | |
| Commercial assay or kit | SAH ELISA kit | Jinma Biotechnology Co. Ltd, Shanghai, China | JS1362-Mu | |
| Commercial assay or kit | HCY ELISA kit | Jinma Biotechnology Co. Ltd, Shanghai, China | JS1363-Mu | |
| Commercial assay or kit | Genistein ELISA kit | Jinma Biotechnology Co. Ltd, Shanghai, China | JS194-SH | |
| Commercial assay or kit | Dibutyl thalate ELISA kit | Jinma Biotechnology Co. Ltd, Shanghai, China | JS291-NC | |
| Commercial assay or kit | CUT&Tag Assay Kit | Vazyme, China | TD903 | |
| Chemical compound, drug | Luzindole | Sigma | L2407 | 5 mg/kg (ip) |
| Chemical compound, drug | SQ22536 | Selleck | S8283 | 2 mg/kg (ip) |
| Chemical compound, drug | Forskolin | Selleck | S2449 | 2 mg/kg (ip) |
| Chemical compound, drug | H89 2HCL | Selleck | S1582 | 5 mg/kg (ip) |
| Chemical compound, drug | Azacitidine | Selleck | S1782 | 10 mg/kg (ip) |
| Chemical compound, drug | 8-Bromo-cAMP | Selleck | S7857 | 5 mg/kg (ip) |
| Chemical compound, drug | Melatonin | Sigma | M5250 | 10 mg/kg (iv) |
| Software, algorithm | GraphPad Prism 9 | GraphPad | | |

