## [Editor Report · eLife Assessment]

This manuscript reports **important** findings on the impact of maternal obesity on offspring metabolism. It presents **solid** evidence that maternal obesity induces genomic methylation alterations in oocytes, which can be partly transmitted to F2 in females, and that melatonin is involved in regulating the hyper-methylation of high fat diet oocytes by increasing the expression of DNMTs via the cAMP/PKA/CREB pathway. This study would be of interest to biologists in the fields of epigenetics and metabolism.

---

## [Referee Report · Joint Public review]

Summary

This manuscript offers significant insights into the impact of maternal obesity on oocyte methylation and its transgenerational effects. Chao and colleagues demonstrated the potential mechanisms behind the DNA methylation changes. The major observations of the work include transgenerational DNA methylation changes in offspring of maternal obesity and metabolites such as methionine and melatonin which correlated with the epigenetic changes. Exogenous melatonin treatment could reverse the effects of obesity. The authors further hypothesized that the linkage may be mediated by the cAMP/PKA/CREB pathway to regulate the expression of DNMTs. This work has done lots of breeding and DNA Methylation analysis across multiple generations, which provides solid data for future research. The results of this work may benefit from deeper data analysis to make more causal analyses and conclusions more concrete.

Strengths

The study employs comprehensive methodologies, including transgenerational breeding experiments, whole genome bisulfite sequencing, and metabolomics analysis, and provides the convincing data.

Weaknesses

The results of this work are correlational, which may require further analysis to establish more concrete conclusions on causal relationships.

---

## [Author Response]

The following is the authors’ response to the original reviews.

**Public Reviews:**

**Reviewer #1 (Public Review):**
With socioeconomic development, more and more people are obese which is an important reason for sub-fertility and infertility. Maternal obesity reduces oocyte quality which may be a reason for the high risk of metabolic diseases for offspring in adulthood. Yet the underlying mechanisms are not well elucidated. Here the authors examined the effects of maternal obesity on oocyte methylation. Hyper-methylation in oocytes was reported by the authors, and the altered methylation in oocytes may be partially transmitted to F2. The authors further explored the association between the metabolome of serum and the altered methylation in oocytes. The authors identified decreased melatonin. Melatonin is involved in regulating the hyper-methylation of high-fat diet (HFD) oocytes, via increasing the expression of DNMTs which is mediated by the cAMP/PKA/CREB pathway.Strengths:This study is interesting and should have significant implications for the understanding of the transgenerational inheritance of GDM in humans.

Thank you for your positive comments to our manuscript.

Weaknesses:The link between altered DNA methylation and offspring metabolic disorders is not well elucidated; how the altered DNA methylation in oocytes escapes reprogramming in transgenerational inheritance is also unclear.

Thanks. These are very good questions. There is a long way to completely elucidate the relationship between methylation and offspring metabolic disorders, and the underlying mechanisms of obtained methylation escaping the reprogramming during development. We would like to explore these in the future.

**Reviewer #2 (Public Review):**
This manuscript offers significant insights into the impact of maternal obesity on oocyte methylation and its transgenerational effects. The study employs comprehensive methodologies, including transgenerational breeding experiments, whole genome bisulfite sequencing, and metabolomics analysis, to explore how high-fat diet (HFD)-induced obesity alters genomic methylation in oocytes and how these changes are inherited by subsequent generations. The findings suggest that maternal obesity induces hyper-methylation in oocytes, which is partly transmitted to F1 and F2 oocytes and livers, potentially contributing to metabolic disorders in offspring. Notably, the study identifies melatonin as a key regulator of this hyper-methylation process, mediated through the cAMP/PKA/CREB pathway.Strengths:The study employs comprehensive methodologies, including transgenerational breeding experiments, whole genome bisulfite sequencing, and metabolomics analysis, and provides convincing data.

Thank you for your positive comments to our manuscript.

Weaknesses:The description in the results section is somewhat verbose. This section (lines 126~227) utilized transgenerational breeding experiments and methylation analysis to demonstrate that maternal obesity-induced alterations in oocyte methylation (including hyper-DMRs and hypo-DMRs) can be partially transmitted to F1 and F2 oocytes and livers. The authors should consider condensing and revising this section for clarity and brevity.

Thanks for your suggestions. We have re-written this parts in the revised manuscript.

There is a contradiction with Reference 3, but the discrepancy is not discussed. In this study, the authors observed an increase in global methylation in oocytes from HFD mice, whereas Reference 3 indicates Stella insufficiency in oocytes from HFD mice. This Stella insufficiency should lead to decreased methylation (Reference 33). There should be a discussion of how this discrepancy can be reconciled with the authors' findings.

Thanks for your suggestions. As reported by Reference 33, STELLA prevents hypermethylation in oocytes by sequestering UHRF1 from the nuclei which recruits DNMT1 into nuclei. Han *et al.* reported that obesity induced by high-fat diet reduces STELLA level in oocytes. These indicate that STELLA insufficiency might induce hypermethylation in oocytes, although significant hypermethylation in obese oocytes is not reported by Han *et al.* using immunofluorescence. This contradiction may be caused by the limited sample sizes (n=14) used by Han *et al.* We have added a brief discussion in the revised manuscript.

**Reviewer #3 (Public Review):**
Summary:Maternal obesity is a health problem for both pregnant women and their offspring. Previous works including work from this group have shown significant DNA methylation changes for offspring of obese pregnancies in mice. In this manuscript, Chao et al digested the potential mechanisms behind the DNA methylation changes. The major observations of the work include transgenerational DNA methylation changes in offspring of maternal obesity, and metabolites such as methionine and melatonin correlated with the above epigenetic changes. Exogenous melatonin treatment could reverse the effects of obesity. The authors further hypothesized that the linkage may be mediated by the cAMP/PKA/CREB pathway to regulate the expression of DNMTs.Strengths:The transgenerational change of DNA methylation following HFD is of great interest for future research to follow. The metabolic treatment that could change the DNA methylation in oocytes is also interesting and has potential relevance to future clinical practice.

Thank you for your positive comments to our manuscript.

Weaknesses:The HFD oocytes have more 5mC signal based on staining and sequencing (Fig 1A-1F). However, the authors also identified almost equal numbers of hyper- and hypo-DMRs, which raises questions regarding where these hypo-DMRs were located and how to interpret their behaviors and functions. These questions are also critical to address in the following mechanistic dissections as the metabolic treatments may also induce bi-directional changes of DNA methylation. The authors should carefully assess these conflicts to make the conclusions solid.

Thanks for the helpful comments and suggestions. As presented in Fig. 1F, there is an increase of methylation level in promoter and exon regions and there is a decrease in intron, utr3 and repeat regions. According to the suggestions, we further analyzed the distribution of DMRs, and found that hypo-DMRs were mainly distributed at utr3, intron, repeat, and tes regions compared with hyper-DMRs (Fig. S3). These suggest that the distribution of DMRs in genome is not random.

The transgenerational epigenetic modifications are controversial. Even for F0 offspring under maternal obesity, there were different observations compared to this work (Hou, YJ., et al. Sci Rep, 2016). The authors should discuss the inconsistencies with previous works.

Thanks for the suggestions. There are contradictions on the whole genome DNA methylation of oocytes in obese mice. Hou YJ *et al.* in 2016 reported that obesity reduces the whole genome DNA methylation of NSN GV oocytes using immunofluorescence. In 2018, Han LS *et al.* reported that the whole genome 5mC of oocytes is not significantly influenced by obesity using immunofluorescence, but they find the *Stella* level is reduced in oocytes by obesity. *Stella* locates in the cytoplasm and nuclei of oocytes and sequesters *Uhrf1* from the nuclei. *Stella* knockout in oocytes results in about twofold increase of global methylation in MII oocytes via recruiting more DNMT1 into nuclei. These suggest that the global methylation of oocytes in obese mice should be increased, but the similar methylation in oocytes between obese and non-obese mice is reported by Han LS *et al*. Thus, the contradiction may be induced by the different sample size in our manuscript and previous studies, and Hou YJ and colleagues just examined the methylation of NSN GV oocytes. As present in Stella+/- oocytes, the global methylation of oocytes is normal, which suggest that the insufficiency of *Stella* may be not the main reason for the increased methylation of oocytes in obese mice. We have added a brief discussion in the revised manuscript.

In addition to the above inconsistencies, the DNA methylation analysis in this work was not carefully evaluated. Several previous works were evaluating the DNA methylation in mice oocytes, which showed global methylation levels of around 50% (Shirane K, et al. PLoS Genet, 2013; Wang L., et al, Cell, 2014). In Figure 1E, the overall methylation level is about 23% in control, which is significantly different from previous works. The authors should provide more details regarding the WGBS procedure, including but not limited to sequencing coverage, bisulfite conversion rate, etc.

Thanks for the good questions. Smallwood *et al.* reported the the CG methylation of MII oocyte is about 33.1% (Smallwood et al. Nature Methods, 2014) using single-cell genome-wide bisulfite sequencing. Shirane K *et al.* reported that the average methylation level of GV oocytes is 37.9%. Kobayashi H *et al.* Reported that the CG methylation in GV oocytes is about 40% (Kobayashi H et al. Plos Genet. 2012). CG methylation in fully grown oocytes is about 38.7% (Maenohara S et al. Plos Genet. 2017). The variation of methylation in oocytes is associated with sequencing methods, sequencing depth, and mapping rates. In the present study, whole genome bisulfite sequencing (WGBS) for small sample and methylation analysis were performed by NovoGene. The reads are 31613641 to 37359643, unique mapping rate is ≥32.88%, conversation rate is > 99.44%, and sequencing depth is 2.45 to 2.75. Relative information is presented in Table S1. The sequencing depth might be a reason for the inconsistence. But we further confirmed our sequencing results using bisulfite sequencing (BS), and the result is similar between BS and WGBS results. These findings suggest that our results are reliable.

**Recommendations for the authors:**

**Reviewer #1 (Recommendations For The Authors):**
(1) Since the results show that melatonin may play a role in hyper-methylation, the authors need to give some basic information in the Introduction section.

Thanks. We added more information in the section of Introduction.

(2) There are many differential metabolites identified. Besides melatonin, other differential metabolites are involved in the altered methylation in oocytes

These is a good question. We firstly filtered the differential metabolites which may be involved in methylation, and then further filtered these metabolites according to the relative DNA methylation pathways and published papers. After that, we confirmed the concentrations of relative metabolites in the serum using ELISA. Certainly, we can not completely exclude all the metabolites which might involved in regulating DNA methylation.

(3) The altered methylation would be found in the F1 tissues. Did the authors examine the other parts besides the liver?

Thank you. In the present study, we didn’t examined the DNA methylation in the other tissues besides the liver. We agree that the altered methylation should be observed in the other tissues.

(4) Did the authors try or guess how many generations the maternal obesity-induced genomic methylation alterations can be transmitted?

Thanks. This is a good question. Takahashi Y and colleagues reported that obtained DNA methylation at CpG island can be transmitted across multiple generations using DNA methylation-edited mouse (Takahashi Y et al. 2023, cell). Similar inheritance is also reported by other studies using different models.

(5) The F2 is indirectly affected by maternal obesity, so the evidence is not enough to prove the transgenerational inheritance of the altered methylation.

Thanks. We find the altered DNA methylation in F2 tissue and oocytes is similar to that in F1 oocytes. These suggest the altered DNA methylation in F2 oocytes should be at least partly transmitted to F3. Previous paper (Takahashi Y et al. 2023, cell) confirms that obtain DNA methylation in CpG island can be transmitted across several generations through paternal and maternal germ lines. Certainly, it’s better if it is examined in F3 tissues.

**Reviewer #2 (Recommendations For The Authors):**
(1) Figure Font Size: The font sizes in the figures are quite inconsistent. Please try to uniform the font size of similar types of text.

Thanks for your suggestions. We re-edited the relative figures in the revised manuscript.

(2) Figure Clarity: Ensure that all critical information in the figures is clearly visible, such as in Figure 3C.

Thank you. We revised this figure.

(3) Figure 1B, C: The position of the asterisks ("**") is not centered in the corresponding columns, and the font size is too small. Please correct this and address similar issues in other figures.

Thank you for your suggestions. We re-edited these in the revised figures.

(4) Line 126: The current expression is confusing. It may be revised to: "Both the oocyte quality and the uterine environment can contribute to adult diseases, which may be mediated by epigenetic modifications."

Thanks. We revised this sentence in the revised manuscript.

(5) Missing Panel in Figure 3: Figure 3 is missing panel 3N.

Thank you so much. We corrected it in the revised manuscript.

(6) Figure Panel Order: Please adjust the order of the panels in the figures to follow a logical reading sequence.

Thank you. We changed the orders in the revised manuscript.

(7) Line 493: Correct "inthe" to "in the".

Thank you. We revised it.

(8) Lines 102-106: Polish the wording and expression, an example as follows: "We analyzed the differentially methylated regions (DMRs) in oocytes from both HFD and CD groups and identified 4,340 DMRs. These DMRs were defined by the criteria: number of CG sites {greater than or equal to} 4 and absolute methylation difference {greater than or equal to} 0.2. Among these, 2,013 were hyper-DMRs (46.38%) and 2,327 were hypo-DMRs (53.62%) (Fig. 1G). These DMRs were distributed across all chromosomes (Fig. 1H). "

Thank you! We re-wrote these parts in the revised manuscript.

**Reviewer #3 (Recommendations For The Authors):**
The sample numbers should be annotated in the figure legend for all the bar plots using Image J. The lines in Figures 2B and 2C were without error bars. How many mice were used for these plots?

Thanks for your suggestions. We added the sample size in the revised manuscript. We made a mistake when we prepared the pictures for figure 2B and figure 2C, which resulted in missing the error bars. We have corrected these pictures. Thanks again!

The authors should revise the panel arrangement of the figures (Figure 2, Figure 5, etc) to make them more clear and readable.

Thank you! We have revised these in the revised manuscript.

The writing should be improved since there were multiple typos and unclear expressions. AI tools like Grammarly or ChatGPT may help.

Thank you! We have re-edited the language in the revised manuscript using AI tools.

Please recheck the immunofluorescence images for clear interpretability. For example, in Figure 5F (H89 treated), the GV is all the way at the edge of the oocyte, and the oocyte in the DIC image appears like it is partially lysed. The DIC images and the DAPI images are not clear enough.

Thanks for your suggestions. We have re-edited these pictures in the revised manuscript.

Another concern is that the Methods describes the immunofluorescence preparation for 5mC and 5hmC staining as a simple fixation in 4% paraformaldehyde followed by permeabilization with .5% TritonX-100, but there is no antigen exposure step described, a step that is normally required for visualizing these DNA modifications (e.g., 4N HCl).

Thanks. Sorry for that we didn’t describe the methods clearly. We have added more information about the methods in the revised manuscript.

The metabolomic analysis revealed a highly significant increase in dibutylphthalate, genistein, and daidzein in the control mice. The presence of these exogenous metabolites suggests that the diets differed in many aspects, not just fat content, so it would be very difficult to interpret the results as related to a high-fat diet alone. Both daidzein and genistein are phytoestrogens and dibutylphthalate is a plasticizer, suggesting differences in the diet and/or in the materials used to collect the samples for analysis from the mice. The Methods define the high-fat diet adequately, as the formulation can be found online using the catalog number. However, the control diet is just listed as "normal diet", so one has no idea what is in it

Thank you for your good questions. The daidzein and genistein may be from the diets and the dibutylthalate may be from the materials used to collect samples. If so, these should be similar between groups. Thus, we added the formulation of normal diet in the revised manuscript. The raw materials of normal diet include corn, bean pulp, fish meal, flour, yeast powder, plant oil, salt, vitamins, and mineral elements. According to the suggestions, we re-checked the data about these metabolites, and found that the abundance of these metabolites was low. And the result of these metabolites was at a low confidence level because the iron of these metabolites was only mapped to ChemSpider(HMDB,KEGG,LIPID MAPS). To further confirm these results, we examined these metabolites in serum using ELISA, and results revealed that the concentrations of genistein and dibutylthalate were similar between groups. These results suggest that these metabolites may be not involved in the altered methylation of oocytes induced by obesity.